# AReUReDi: Annealed Rectified Updates for Refining Discrete Flows with Multi-Objective Guidance

**Tong Chen,**[1] **Yinuo Zhang,**[2] **Pranam Chatterjee**[1,2,3,†]

[1]Department of Computer and Information Science, University of Pennsylvania
[2]Centre for Computational Biology, Duke-NUS Medical School, Singapore
[3]Department of Bioengineering, University of Pennsylvania

[†]Corresponding author: pranam@seas.upenn.edu

## Abstract

Designing sequences that satisfy multiple, often conflicting, objectives is a central challenge in therapeutic and biomolecular engineering. Existing generative frameworks largely operate in continuous spaces with single-objective guidance, while discrete approaches lack guarantees for multi-objective Pareto optimality. We introduce **AReUReDi** (**A**nnealed **Re**ctified **U**pdates for **Re**fining **Di**screte Flows), a discrete optimization algorithm with theoretical guarantees of convergence to the Pareto front. Building on Rectified Discrete Flows (ReDi), AReUReDi combines Tchebycheff scalarization, locally balanced proposals, and annealed Metropolis-Hastings updates to bias sampling toward Pareto-optimal states while preserving distributional invariance. Applied to peptide and SMILES sequence design, AReUReDi simultaneously optimizes up to five therapeutic properties (including affinity, solubility, hemolysis, half-life, and non-fouling) and outperforms both evolutionary and diffusion-based baselines. These results establish AReUReDi as a powerful, sequence-based framework for multi-property biomolecule generation.

## 1 Introduction

The design of biological sequences must account for multiple, often conflicting, objectives (Naseri & Koffas, 2020). Therapeutic molecules, for example, must combine high binding affinity with low immunogenicity and favorable pharmacokinetics (Tominaga et al., 2024); CRISPR guide RNAs require both high on-target activity and minimal off-target effects (Mohr et al., 2016; Schmidt et al., 2025); and synthetic promoters must deliver strong expression while remaining tissue-specific (Artemyev et al., 2024). These examples illustrate that biomolecular engineering is inherently a multi-objective optimization problem.

Yet, most computational frameworks continue to optimize single objectives in isolation (Zhou et al., 2019; Nehdi et al., 2020; Nisonoff et al., 2025). While such approaches can reduce toxicity (Kreiser et al., 2020; Sharma et al., 2022) or improve thermostability (Komp et al., 2025), they often create adverse trade-offs: high-affinity peptides may be insoluble or hemolytic, and stabilized proteins may lose specificity (Bigi et al., 2023; Rinauro et al., 2024). Black-box multi-objective optimization (MOO) methods such as evolutionary search and Bayesian optimization have long been applied to molecular design (Zitzler & Thiele, 1998; Deb, 2011; Ueno et al., 2016; Frisby & Langmead, 2021), but these approaches scale poorly in high-dimensional sequence spaces.

To overcome this, recent generative approaches have incorporated controllable multi-objective sampling (Li et al., 2018; Sousa et al., 2021; Yao et al., 2024). For instance, ParetoFlow (Yuan et al., 2024) leverages continuous-space flow matching to generate Pareto-optimal samples. However, extending such guarantees to biological sequences is challenging, since discrete data typically require embedding into continuous manifolds, which distorts token-level structure and complicates property-based guidance (Beliakov & Lim, 2007; Michael et al., 2024).

A more direct path lies in discrete flow models (Campbell et al., 2024; Gat et al., 2024; Dunn & Koes, 2024). These models define probability paths over categorical state spaces, either through simplex-based interpolations (Stark et al., 2024; Davis et al., 2024; Tang et al., 2025a) or jump-process flows that learn token-level transition rates (Campbell et al., 2024; Gat et al., 2024). Recent advances have shown their promise for controllable single-objective generation (Nisonoff et al., 2025; Tang et al., 2025a), but no framework yet achieves Pareto guidance across multiple objectives.

Here, the notion of rectification provides a crucial building block. In the continuous setting, *Rectified Flows* (Liu et al., 2023) learn to straighten ODE paths between distributions, thereby reducing convex transport costs and enabling efficient few-step or even one-step sampling. Recently, **ReDi** (*Rectified Discrete Flows*) (Yoo et al., 2025) extended this principle to discrete domains. By iteratively refining the coupling between source and target distributions, ReDi provably reduces factorization error (quantified as conditional total correlation) while maintaining distributional fidelity. This makes ReDi highly effective for efficient discrete sequence generation. However, ReDi does not address the multi-objective setting, as it lacks a mechanism to steer sampling toward the *Pareto front*, where improvements in one objective cannot be made without degrading another. This is a critical limitation for biomolecular design, where trade-offs define practical success.

To address this, we introduce **AReUReDi** (**A**nnealed **Re**ctified **U**pdates for **Re**fining **Di**screte Flows), a new framework that extends rectified discrete flows with multi-objective guidance. AReUReDi integrates three innovations: (i) *annealed Tchebycheff scalarization*, which gradually sharpens the focus on balanced solutions across objectives (Lin et al., 2024a); (ii) *locally balanced proposals*, which combine the generative prior of ReDi with multi-objective guidance while ensuring reversibility; and (iii) *Metropolis-Hastings updates*, which preserve exact distributional invariance and guarantee convergence to Pareto-optimal states. Together, these mechanisms refine rectified discrete flows into a principled Pareto sampler.

Our key contributions are:

1. We propose AReUReDi, the first multi-objective extension of rectified discrete flows, integrating annealed scalarization, locally balanced proposals, and MCMC updates.

2. We provide theoretical guarantees that AReUReDi preserves distributional invariance and converges to the Pareto front with full coverage.

3. We demonstrate that AReUReDi can optimize up to five competing biological properties simultaneously, including affinity, solubility, hemolysis, half-life, and non-fouling.

4. We benchmark AReUReDi against classical MOO algorithms and state-of-the-art discrete diffusion approaches, showing superior trade-off navigation and biologically plausible sequence designs.

A detailed discussion of Related Work is provided in Appendix Section A.

## 2 PRELIMINARIES

### 2.1 DISCRETE FLOW MATCHING

Let $\mathcal{S} = V^L$ denote the discrete state space, where $V$ is a vocabulary of size $K$ and each $x = (x_1, \dots, x_L) \in \mathcal{S}$ is a sequence of tokens. A *discrete flow matching (DFM)* model (Campbell et al., 2024; Gat et al., 2024; Dunn & Koes, 2024) defines a probability path $\{p_t\}_{t \in [0,1]}$ interpolating between a simple source distribution $p_0$ and a target distribution $p_1$ by means of a coupling $\pi(x_0, x_1)$ and conditional bridge distributions $p_t(x_t \mid x_0, x_1)$. The model is trained to approximate conditional transitions $p_{s|t}(x_s \mid x_t)$ for $0 \le t < s \le 1$.

Since the joint distribution over $L$ coordinates is intractable, DFMs employ a factorization

$$p_{s|t}(x_s \mid x_t) \approx \prod_{i=1}^{L} p_{s|t}\left(x_s^i \mid x_t\right),$$

which introduces a discrepancy measured by the conditional total correlation

$$\text{TC}_{s|t} \;=\; \text{KL}\!\left(p_{s|t}(x_s \mid x_t) \,\Big\|\, \prod_{i=1}^{L} p_{s|t}(x_s^i \mid x_t)\right).$$

This quantity captures the inter-dimensional dependencies neglected under factorization, and grows with larger step sizes (Stark et al., 2024; Davis et al., 2024; Tang et al., 2025a). As a result, DFMs are accurate in the many-step regime but degrade under few-step or one-step generation.

## 2.2 RECTIFIED DISCRETE FLOW

To mitigate factorization error, **Rectified Discrete Flow (ReDi)** (Yoo et al., 2025) introduces an iterative rectification of the coupling $\pi$. Starting from an initial coupling $\pi^{(0)}(x_0, x_1)$, a DFM is trained under $\pi^{(k)}$ to produce new source–target pairs, defining an empirical joint distribution $\hat{\pi}^{(k)}$. The coupling is then updated via

$$\pi^{(k+1)}(x_0, x_1) \;\propto\; \pi^{(k)}(x_0, x_1)\,\frac{p_{\theta^{(k)}}(x_1 \mid x_0)}{p_{\theta^{(k)}}(x_1)},$$

where $p_{\theta^{(k)}}(x_1 \mid x_0)$ is the conditional distribution learned at iteration $k$. This yields a sequence of couplings $\{\pi^{(k)}\}_{k \geq 0}$ with provably decreasing conditional TC,

$$\text{TC}_{s|t}(\pi^{(k+1)}) \;\leq\; \text{TC}_{s|t}(\pi^{(k)}).$$

By progressively reducing factorization error, ReDi produces a well-calibrated base distribution $p_1$ with low inter-dimensional correlation. This base distribution provides reliable marginal transition probabilities $p_t^i(\cdot \mid x_t)$ for each coordinate $i$ at time $t$, which serve as the generative prior in the AReUReDi framework. Rectification follows the same principle as *Rectified Flow* in continuous domains (Liu et al., 2023), where iterative refinement straightens ODE paths and decreases transport costs.

# 3 AReUReDi: ANNEALED RECTIFIED UPDATES FOR REFINING DISCRETE FLOWS

With an efficient discrete flow-based generation framework in hand, we develop AReUReDi that extends ReDi (Yoo et al., 2025) to the multi-objective optimization setting, where the goal is to generate discrete samples that approximate the Pareto front of multiple competing objectives. Starting from a pre-trained ReDi model, AReUReDi incorporates annealed guidance, locally balanced proposals, and Metropolis-Hastings updates to progressively bias the sampling process toward Pareto-optimal states while preserving the probabilistic guarantees of the underlying flow (Algorithm 1).

## 3.1 PROBLEM SETUP

Let the discrete search space be $\mathcal{S} = \mathcal{V}^L$, where $\mathcal{V}$ is a finite vocabulary of size $K$ and each state $x = (x_1, \ldots, x_L) \in \mathcal{S}$ is a sequence of tokens. We assume access to a pre-trained ReDi model that provides marginal transition probabilities $p_t^i(\cdot \mid x_t)$ for each position $i$ and time $t$. In addition, we are given $N$ pre-trained scalar objective functions $s_n : \mathcal{S} \to \mathbb{R}$, where $n = 1, \ldots, N$, and $\tilde{s}_n(x)$ are their normalized counterparts with outputs mapped to $[0, 1]$ to support balanced updates for each objective. The sampling task is to construct a Markov chain whose stationary distribution concentrates on states that approximate the Pareto front of the normalized objectives $\tilde{s}_1, \ldots, \tilde{s}_N$.

## 3.2 ANNEALED MULTI-OBJECTIVE GUIDANCE

To direct sampling toward the Pareto front, AReUReDi introduces a scalarized reward

$$S_\omega(x) = \min_{1 \leq n \leq N} \omega_n \, \tilde{s}_n(x),$$

where the weight vector $\omega = [\omega_1, \ldots, \omega_N]$ lies in the probability simplex $\Delta^{N-1}$ and balances the different objectives. This Tchebycheff scalarization promotes solutions that are simultaneously strong

across all objectives rather than excelling in only a subset (Miettinen, 1999). The scalarized reward is converted into a guidance weight

$$W_{\eta_t,\omega}(x) = \exp\left(\eta_t S_\omega(x)\right),$$

where the parameter $\eta_t > 0$ controls the strength of the guidance at each iteration $t$. AReUReDi incorporates an annealing schedule for $\eta_t$:

$$\eta_t \ = \ \eta_{\min} + \left(\eta_{\max} - \eta_{\min}\right)\frac{t}{T-1},$$

so that the chain begins with a small value of $\eta_t$ to encourage wide exploration of the state space and gradually increases $\eta_t$ to focus sampling on high-quality Pareto candidates. This annealing strategy mirrors simulated annealing but operates directly on the scalarized objectives within the discrete flow framework.

## 3.3 Locally Balanced Proposals

Given the current state $x_t$, AReUReDi updates one coordinate $i \in \{1, \ldots, L\}$ at a time using a locally balanced proposal that blends the generative prior of ReDi with the multi-objective guidance. First, a candidate set of replacement tokens is drawn from the ReDi marginal $p_t^i(\cdot \mid x_t)$, optionally pruned using top-p to retain only the most promising alternatives for computational efficiency. For each candidate token $y$, the algorithm computes the ratio

$$r_i(y; x_t) = \frac{W_{\eta_t,\omega}\left(x_t^{(i\leftarrow y)}\right)}{W_{\eta_t,\omega}(x_t)},$$

which measures the change in scalarized reward if $x_t^i$ were replaced by $y$. The ratio $r_i(y; x_t)$ is then transformed by a balancing function $g : \mathbb{R}_+ \to \mathbb{R}_+$ that satisfies the symmetry condition $g(u) = u\,g(1/u)$. Typical choices include Barker's function $g(u) = \frac{u}{1+u}$ and the square-root function $g(u) = \sqrt{u}$. This symmetry ensures that the resulting Markov chain admits the desired stationary distribution. Using the balanced function, the unnormalized proposal for a candidate token $y$ takes the form

$$\tilde{q}_i(y \mid x_t) = p_t^i(y \mid x_t)\,g\left(r_i(y; x_t)\right),$$

which is then normalized over the candidate set to yield the final proposal distribution $q_i(y \mid x_t)$. This construction allows the proposal to favor states with higher scalarized reward while remaining reversible with respect to the target distribution.

## 3.4 Metropolis-Hastings Update

A candidate token $y^\star$ is drawn from the final proposal distribution $q_i(\cdot \mid x_t)$ and forms the proposed state $x_{\text{prop}} = x_t^{(i\leftarrow y^\star)}$. The proposal is accepted with the standard Metropolis-Hastings probability (Hastings, 1970)

$$\alpha_i(x_t, x_{\text{prop}}) = \min\left\{1, \frac{\pi_{\eta_t,\omega}(x_{\text{prop}})\,q_i(x_t^i \mid x_{\text{prop}})}{\pi_{\eta_t,\omega}(x_t)\,q_i(y^\star \mid x_t)}\right\},$$

where we define $\pi_{\eta_t,\omega}(x) \ \propto \ p_1(x)\,W_{\eta_t,\omega}(x) \ = \ p_1(x)\,\exp\left(\eta_t S_\omega(x)\right)$. With Barker's balancing function, the acceptance probability simplifies to one, ensuring automatic acceptance of proposals and faster mixing. Other choices, such as the square-root function, trade higher acceptance rates for more conservative moves.

The annealed, locally balanced updates are repeated for $T$ iterations and end with the final sample $x_1$ whose objective scores are jointly optimized. Building on ReDi's well-calibrated base distribution with low inter-dimensional correlation, AReUReDi biases this base toward Pareto-optimal regions while maintaining support over the full state space. With the standard Metropolis-Hastings acceptance rule, the resulting kernel preserves the reward-tilted target distribution, yielding distributional invariance and asymptotic Pareto-coverage guarantees (Appendix B). We evaluate a monotone-accept variant for finite-budget efficiency in Section 4.

Table 1: AReUReDi generates wild-type peptide binders for 8 diverse protein targets, optimizing five therapeutic properties: hemolysis, non-fouling, solubility, half-life (in hours), and binding affinity. Each value represents the average of 100 AReUReDi-designed binders.

| Name | Binder Length | Hemolysis | Non-Fouling | Solubility | Half-Life (h) | Affinity |
|---|---|---|---|---|---|---|
| AMHR2 | 8 | 0.9156 | 0.8613 | 0.8564 | 45.73 | 7.0608 |
| AMHR2 | 12 | 0.9384 | 0.8872 | 0.8810 | 52.52 | 7.2284 |
| AMHR2 | 16 | 0.9420 | 0.8914 | 0.8755 | 63.34 | 7.2533 |
| EWS::FLI1 | 8 | 0.9186 | 0.8630 | 0.8619 | 44.77 | 5.8424 |
| EWS::FLI1 | 12 | 0.9345 | 0.8819 | 0.8796 | 59.11 | 6.2007 |
| EWS::FLI1 | 16 | 0.9416 | 0.8875 | 0.8807 | 64.32 | 6.4195 |
| MYC | 8 | 0.9180 | 0.8627 | 0.8627 | 44.13 | 6.4082 |
| OX1R | 10 | 0.9302 | 0.8687 | 0.8563 | 50.14 | 7.1882 |
| DUSP12 | 9 | 0.9240 | 0.8669 | 0.8633 | 48.14 | 6.1276 |
| 1B8Q | 8 | 0.9214 | 0.8680 | 0.8654 | 42.63 | 5.7130 |
| 5AZ8 | 11 | 0.9293 | 0.8732 | 0.8605 | 58.33 | 6.2792 |
| 7JVS | 11 | 0.9313 | 0.8840 | 0.8743 | 56.49 | 6.8449 |

# 4 EXPERIMENTS

To the best of our knowledge, no public datasets exist for benchmarking multi-objective optimization algorithms on biological sequences. We therefore developed two benchmarks to evaluate AReUReDi, focusing on the generation of wild-type peptide sequences and chemically-modified peptide SMILES. These tasks are supported by two core components: the generative models described in Appendix C and the objective-scoring models validated in Appendix F. Leveraging these models, we demonstrate AReUReDi's efficacy on a wide range of tasks and examples.

Although AReUReDi provides theoretical guarantees (Appendix B) for the *unconstrained* MH sampler, these guarantees are asymptotic and may require many sampling steps to manifest. To improve sampling efficiency under a fixed budget, we use a monotone-accept heuristic in all reported experiments, accepting a proposed token update only if it increases the weighted-sum utility. This heuristic makes the resulting Markov chain non-reversible, so the exact invariance and Pareto-coverage guarantees in Appendix B do not strictly apply. Nonetheless, it consistently improves finite-budget optimization performance (Table S4).

## 4.1 AREUREDI EFFECTIVELY BALANCES EACH OBJECTIVE TRADE-OFF

With pre-trained PepReDi in hand, we first focus on validating AReUReDi's capability of balancing multiple conflicting objectives. We performed two sets of experiments for wild-type peptide binder generation with three property guidance, and in ablation experiment settings, we removed one or more objectives. In the binder design task for target 7LUL (hemolysis, solubility, affinity guidance; Table S5), omitting any single guidance causes a collapse in that property, while the remaining guided metrics may modestly improve. Likewise, in the binder design task for target CLK1 (affinity, non-fouling, half-life guidance; Table S6), disabling non-fouling guidance allows half-life to exceed 96 hours but drives non-fouling near zero, and disabling half-life guidance preserves non-fouling yet reduces half-life below 2 hours. In contrast, enabling all guidance signals produces the most balanced profiles across all objectives. These results confirm that AReUReDi precisely targets chosen objectives while preserving the flexibility to navigate conflicting objectives and push samples toward the Pareto front.

## 4.2 AREUREDI GENERATES WILD-TYPE PEPTIDE BINDERS UNDER FIVE PROPERTY GUIDANCE

We next benchmark AReUReDi on a wild-type peptide binder generation task guided by five different properties that are critical for therapeutic discovery: hemolysis, non-fouling, solubility, half-life, and binding affinity. To evaluate AReUReDi in a controlled setting, we designed 100 peptide binders per target for 8 diverse proteins, structured targets with known binders (3IDJ, 5AZ8, 7JVS), structured targets without known binders (AMHR2, OX1R, DUSP12), and intrinsically disordered targets (EWS::FLI1, MYC) (Table 1). Across all targets and across multiple binder lengths, the

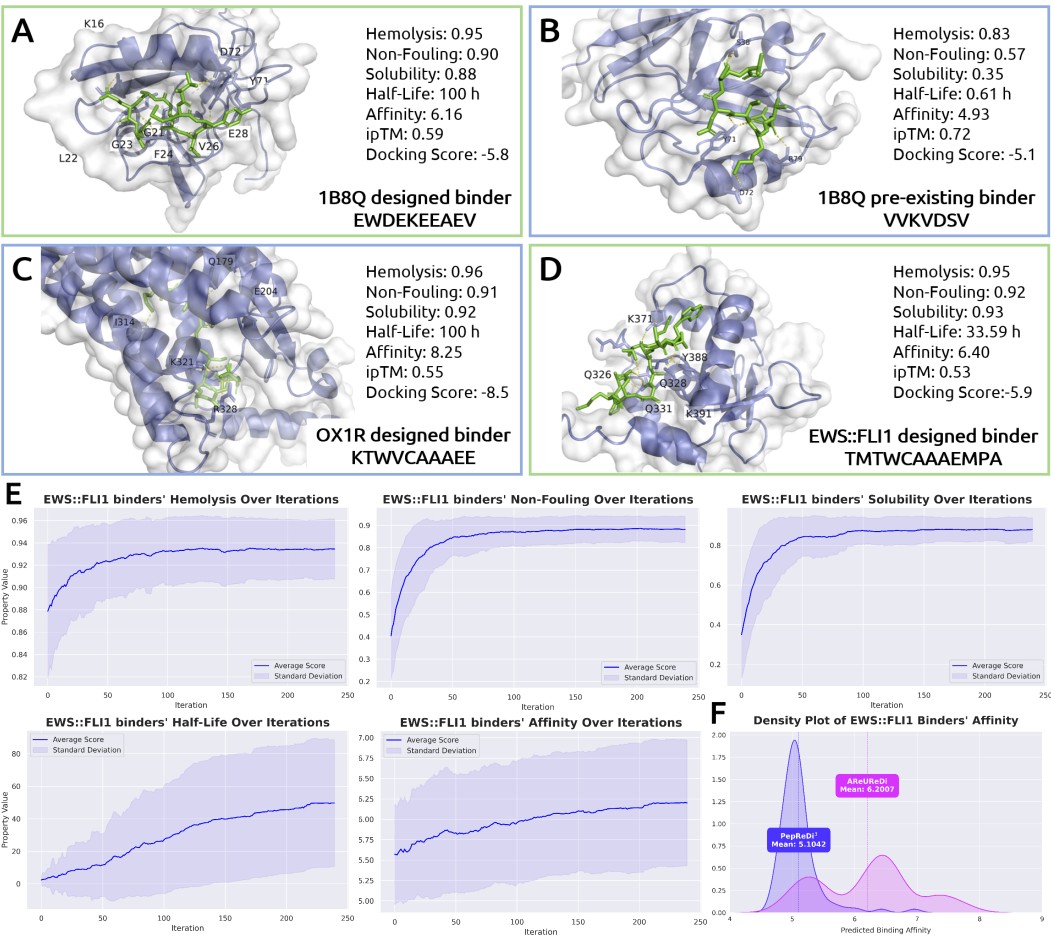

Figure 1: **(A), (B)** Complex structures of PDB 1B8Q with an AReUReDi-designed binder and its pre-existing binder. **(C), (D)** Complex structures of OX1R and EWS::FLI1 with an AReUReDi-designed binder. Five property scores are shown for each binder, along with the ipTM score from AlphaFold3 and docking score from AutoDock VINA. Interacting residues on the target are visualized. **(E)** Plots showing the mean scores for each property across the number of iterations during AReUReDi's design of binders of length 12-aa for EWS::FLI1. **(F)** A density plot illustrating the distribution of predicted property scores for AReUReDi-designed EWS::FLI1 binders of length 12-aa, compared to the peptides generated unconditionally by PepReDi[3].

generated peptides achieve superior hemolysis rates (0.91-0.94), high non-fouling (>0.86) and solubility (>0.85), extended half-life (42-64 h), and strong affinity scores (5.7-7.3), demonstrating both balanced optimization and robustness to sequence length.

For the target proteins with pre-existing binders, we compared the property values between their known binders with AReUReDi-designed ones (Figure 1A,B, S1). The designed binders significantly outperform the pre-existing binders across all properties without compromising the binding potential, which is further confirmed by the ipTM scores computed by AlphaFold3 (Abramson et al., 2024) and docking scores calculated by AutoDock VINA (Trott & Olson, 2010). Although the AReUReDi-designed binders bind to similar target positions as the pre-existing ones, they differ significantly in sequence and structure, demonstrating AReUReDi's capacity to explore the vast sequence space for optimal designs. For target proteins without known binders, complex structures were visualized using one of the AReUReDi-designed binders (Figure S2). The corresponding property scores, as well as ipTM and docking scores, are also displayed. Some of the designed binders showed longer half-life, while others excelled in non-fouling and solubility, underscoring the comprehensive exploration of the sequence space by AReUReDi.

To evaluate our guided generation strategy, we tracked the mean and standard deviation of five property scores across 100 generated binders (length 12) targeting EWS::FLI1 at each iteration (Figure 1E).

Table 2: AReUReDi outperforms traditional multi-objective optimization algorithms in designing wild-type peptide binders guided by five objectives. Each value represents the average of 100 designed binders. The table also records the average runtime for each algorithm to design a single binder. The best result for each metric is highlighted in bold.

| Target | Method | Time (s) | Hemolysis | Non-Fouling | Solubility | Half-Life (h) | Affinity |
|--------|--------|----------|-----------|-------------|------------|---------------|----------|
| 1B8Q | MOPSO | 8.54 | 0.8934 | 0.4763 | 0.4684 | 4.45 | 6.0594 |
| | NSGA-III | 33.13 | 0.9138 | 0.5715 | 0.5825 | 7.32 | 7.2178 |
| | SMS-EMOA | 8.21 | 0.8804 | 0.3450 | 0.3511 | 3.02 | 5.955 |
| | SPEA2 | 17.48 | 0.9181 | 0.4973 | 0.5057 | 4.13 | **7.3240** |
| | PepTune + DPLM | **2.46** | 0.8547 | 0.3085 | 0.3213 | 1.17 | 5.2398 |
| | **AReUReDi** | 55 | **0.9214** | **0.8680** | **0.8654** | **22.93** | 5.7130 |
| PPP5 | MOPSO | 11.34 | 0.9117 | 0.4711 | 0.4255 | 1.77 | 6.6958 |
| | NSGA-III | 37.30 | **0.9521** | 0.7138 | 0.7066 | 2.90 | 7.3789 |
| | SMS-EMOA | 8.43 | 0.8758 | 0.4269 | 0.4334 | 1.03 | 6.2854 |
| | SPEA2 | 19.02 | 0.9445 | 0.6221 | 0.6098 | 2.61 | **7.6253** |
| | PepTune + DPLM | **4.80** | 0.8816 | 0.2752 | 0.2636 | 1.27 | 5.8454 |
| | **AReUReDi** | 195 | 0.9412 | **0.896** | **0.8832** | **38.28** | 6.7186 |

All five properties steadily improved, with average scores for solubility and non-fouling properties increasing markedly from around 0.4 to 0.9. The large standard deviation observed in the final half-life and binding affinity values reflects this property's high sensitivity to guidance, as AReUReDi balances the trade-offs between multiple conflicting objectives. We further visualized AReUReDi's impact by comparing the property distribution of the 100 guided peptides to that of 100 peptides unconditionally sampled from PepReDi[3]. The results show that AReUReDi effectively shifted the distribution towards peptides with higher binding affinity. Collectively, these findings demonstrate AReUReDi's capability to steer generation toward simultaneous multi-property optimization.

We benchmarked AReUReDi against four established multi-objective optimization (MOO) baselines (NSGA-III (Deb & Jain, 2013), SMS-EMOA (Beume et al., 2007), SPEA2 (Zitzler et al., 2001), and MOPSO (Coello & Lechuga, 2002)) on two protein targets: 1B8Q, a small protein with known peptide binders (Zhang et al., 1999), and PPP5, a larger protein without characterized binders (Yang et al., 2004) (Table 2). Each method generated 100 candidate binders optimized for five properties: hemolysis, non-fouling, solubility, half-life, and binding affinity. While AReUReDi required longer runtimes than evolutionary baselines, it consistently produced the best trade-offs. For both targets, it designed targets with top hemolysis scores, increased non-fouling and solubility by 30-50%, maintained competitive binding affinity, and even extended the half-life by a factor of 3-13 relative to the next-best method. These results underscore AReUReDi's effectiveness in navigating high-dimensional property landscapes to yield peptide binders with balanced, optimized profiles.

We also compared against PepTune (Tang et al., 2025b), a recent masked discrete diffusion model for peptide design that couples generation with Monte Carlo Tree Search for MOO. PepTune's backbone was adapted to the existing DPLM model (Wang et al., 2024) for wild-type peptide sequence generation. Despite longer runtimes, AReUReDi substantially outperformed PepTune across all objectives, yielding nearly threefold improvements in non-fouling and solubility and a 22-fold increase in half-life. Together, these comparisons demonstrate that AReUReDi surpasses not only traditional MOO algorithms but also the current state-of-the-art diffusion-based approach for multi-objective-guided wild-type peptide binder design.

Since AReUReDi requires more computation than PepTune to design the same number of binders, we compare both methods under a matched wall-clock budget (Table S9). Specifically, the time PepTune needs to generate 100 binders approximately matches the time AReUReDi needs to generate four 8-mer binders for 1B8Q and three 16-mer binders for PPP5. For both tasks, the top-2 AReUReDi binders achieve substantially higher non-fouling, solubility, and half-life, while maintaining comparable hemolysis and affinity. This comparison shows that AReUReDi produces better multi-objective trade-offs, even when PepTune is allowed a much larger candidate pool under the same time budget.

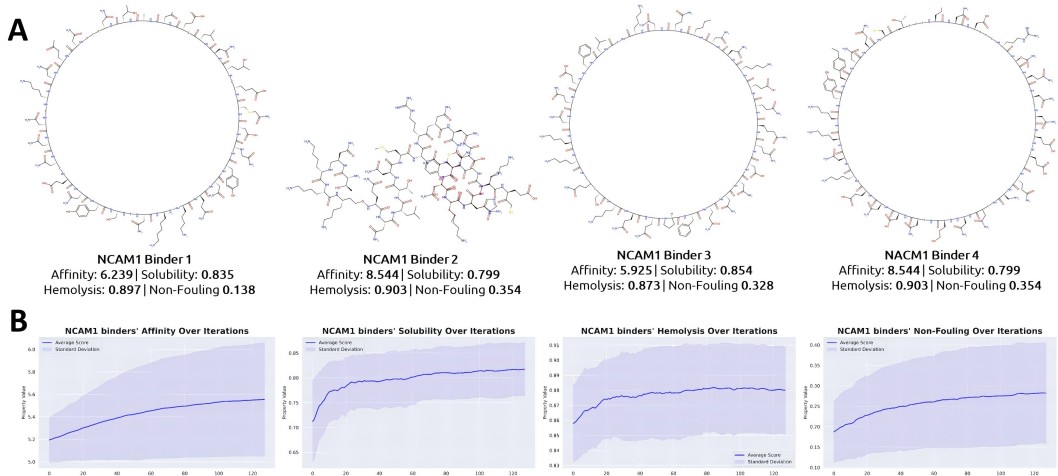

Figure 2: **(A)** Example 2D SMILES structure of AReUReDi-designed peptide binders with four property scores. **(B)** Plots showing the mean scores for each property across the number of iterations during AReUReDi's design of binders of length 200 for NCAM1.

### 4.3 AReUReDi generates therapeutic peptide SMILES under four property guidance

To demonstrate the broad applicability of AReUReDi for multi-objective guided generation of biological sequences, we employed the rectified SMILESReDi model to design chemically-modified peptide binder SMILES sequences for five diverse therapeutic targets. These included the metabolic hormone receptor Glucagon-like peptide-1 receptor (GLP1), the iron transport protein Transferrin receptor (TfR), the Neural Cell Adhesion Molecule 1 (NCAM1), the neurotransmitter transporter GLAST, and the developmental Anti-Müllerian Hormone Receptor Type 2 (AMHR2). For each target, sequence generation was jointly conditioned on a predicted binding-affinity score to the target protein, as long as hemolysis, solubility, and non-fouling, to ensure both potency and desirable physicochemical profiles. Although PepTune is also able to perform multi-property guided design of peptide-binder SMILES sequences, it does not report average property scores for its generated binders, making a direct quantitative comparison with AReUReDi infeasible (Tang et al., 2025b).

We selected and visualized representative binders with the highest predicted binding affinities for each target (Figure 2A, S3A,C, S4A,C). All selected binders achieved high scores across hemolysis, solubility, non-fouling, and binding affinity. During generation, we recorded the mean and standard deviation of all four property scores over 100 binders at each iteration to assess the effectiveness of the multi-objective guidance (Figure 2B, S3B,D, S4B,D). Across all targets, binding affinity scores and non-fouling scores showed steady upward trends throughout the generation process, while hemolysis and solubility scores fluctuated, indicating AReUReDi's effort to balance the four conflicting objectives. Moreover, AReUReDi produces valid sequences with substantially higher diversity and lower SNN than PepTune, indicating both superior novelty and structural variability (Table S3). These findings highlight the versatility and reliability of AReUReDi for the *de novo* design of chemically modified peptide binders across a wide range of therapeutic targets.

A detailed discussion of ablation studies is provided in Appendix Section H.

## 5 Discussion

In this work, we have presented **AReUReDi**, a multi-objective optimization framework that extends rectified discrete flows to generate biomolecular sequences satisfying multiple, often conflicting, properties. By integrating annealed Tchebycheff scalarization, locally balanced proposals, and Metropolis-Hastings updates, AReUReDi provides theoretical guarantees of convergence to the Pareto front while maintaining full coverage of the solution space. Built on high-quality base generators such as PepReDi and SMILESReDi, the method demonstrates broad applicability across amino acid

sequences and chemically modified peptide SMILES. Superior *in silico* results establish AReUReDi as a general, theoretically-grounded tool for multi-property-guided biomolecular sequence design.

While AReUReDi excels in domains like wild-type and chemically-modified peptide designs, future work will extend to other biological modalities, including DNA, RNA, antibodies, and combinatorial genotype libraries, where multi-objective trade-offs are central. From a theoretical perspective, improving AReUReDi's efficiency while maintaining the Pareto convergence guarantees and incorporating uncertainty-aware or feedback-driven guidance remain key directions to explore. Ultimately, AReUReDi provides a foundation for designing the next generation of therapeutic molecules that are not only potent but also explicitly optimized for the diverse properties required for clinical success.

## MEANINGFULNESS STATEMENT

AReUReDi advances our ability to learn meaningful representations of life by turning multi-objective biological constraints into a structured learning signal for discrete sequence models. By guiding rectified discrete flows toward Pareto-balanced solutions, it exposes how fitness trade-offs (e.g., binding, stability, solubility, toxicity proxies) shape the geometry of peptide sequence space and which edits preserve function while improving developability. The framework also yields diverse, constrained samples that can be used to probe representation quality: whether learned features capture biologically relevant variation and support controllable generation. Overall, AReUReDi links generative modeling to interpretable, testable structure in biomolecular landscapes.

## REPRODUCIBILITY STATEMENT

We ensure reproducibility through detailed theoretical, algorithmic, and experimental descriptions of AReUReDi. The complete procedure is formally described in the main text with proofs of convergence guarantees, including the rectified discrete flow foundation, annealed Tchebycheff scalarization, locally balanced proposals, and Metropolis-Hastings updates. Architectures, training details, and datasets for all base generators (PepReDi, SMILESReDi, and PepDFM) are reported with quantitative metrics in the Results and Appendix. Hyperparameter settings, annealing schedules, and sensitivity analyses are provided to facilitate replication, and ablation studies are included to assess the impact of key design choices. Benchmark comparisons against classical multi-objective optimization baselines and diffusion-based methods are tabulated for reference. All datasets used in this work (PepNN, BioLip2, PPIRef, peptide property datasets, and peptide SMILES collections) are publicly available. We will release code, pretrained checkpoints, and sampling scripts for AReUReDi to enable full reproducibility.

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

# Appendix

## A    RELATED WORKS

**Online Multi-Objective Optimization.** Recent work in multi-objective guided generation has focused on online or sequential decision-making, where solutions are refined with new data (Gruver et al., 2023; Jain et al., 2023; Stanton et al., 2022; Ahmadianshalchi et al., 2024). A common approach is Bayesian optimization (BO), which builds a surrogate model and proposes evaluations via acquisition functions (Yu et al., 2020; Shahriari et al., 2015). Multi-objective BO often uses advanced criteria such as EHVI (Emmerich & Klinkenberg, 2008), information gain (Belakaria et al., 2021), or scalarization (Knowles, 2006; Zhang & Li, 2007; Paria et al., 2020). While AReUReDi also employs Tchebycheff scalarization, it operates in an offline setting, where each sequence requires costly evaluation. This contrasts with the sequential, feedback-driven nature of online methods, making direct comparison inappropriate.

**Tchebycheff Scalarization.** Tchebycheff scalarization can identify any Pareto-optimal point and is widely used in multi-objective optimization (Miettinen, 1999). Recent variants include smooth scalarization for gradient-based algorithms (Lin et al., 2024b) and OMD-TCH for online learning (Liu et al., 2024). AReUReDi is, to our knowledge, the first to apply Tchebycheff scalarization for offline generative design of discrete therapeutic sequences. Future work may extend to many-objective problems or alternative utility functions (Lin et al., 2024a; Tu et al., 2023).

**Diffusion and Flow Matching.** Generative approaches such as ParetoFlow and PGD-MOO adapt flow matching or diffusion models for multi-objective optimization (Yuan et al., 2024; Annadani et al., 2025). These operate in continuous or latent spaces, whereas AReUReDi is designed for discrete token spaces inherent to biological sequences. This domain mismatch precludes direct benchmarking.

**Biomolecule Generation.** Offline multi-objective frameworks such as EGD and MUDM have optimized molecules with multiple properties (Sun et al., 2025; Han et al., 2023), but these emphasize 3D structural representations. By contrast, AReUReDi is sequence-only, operating directly over amino acids or SMILES, which makes structural methods unsuitable as direct comparators.

## B    THEORETICAL GUARANTEES

In this section, we establish that AReUReDi converges to Pareto-optimal solutions while preserving coverage of the entire Pareto front. We assume throughout that the state space $\mathcal{S}$ is finite, all objective functions $s_n$ are bounded, and their normalized versions $\tilde{s}_n$ map to $[0, 1]$.

### B.1    PRELIMINARY DEFINITIONS

**Definition (Pareto Optimality).** A state $x^* \in \mathcal{S}$ is *Pareto optimal* if there exists no $y \in \mathcal{S}$ such that $\tilde{s}_n(y) \geq \tilde{s}_n(x^*)$ for all $n \in \{1, \ldots, N\}$ with strict inequality for at least one $n$.

**Definition (Pareto Front).** The Pareto front is $\mathcal{P} = \{x \in \mathcal{S} : x \text{ is Pareto optimal}\}$.

**Definition (Interior Weight Vector).** A weight vector $\omega \in \Delta^{N-1}$ is *interior* if $\omega_n > 0$ for all $n$.

### B.2    MAIN THEORETICAL RESULTS

**Theorem (Invariance).** The Markov kernel defined by the Locally Balanced Proposal (LBP) and Metropolis–Hastings update leaves the distribution

$$\pi_{\eta,\omega}(x) \propto p_1(x) \exp\big(\eta S_\omega(x)\big)$$

invariant for every guidance strength $\eta > 0$ and weight vector $\omega \in \Delta^{N-1}$.

*Proof.* We prove this in two steps: first showing that single-coordinate updates preserve detailed balance, then that random-scan mixtures preserve invariance.

**Step 1: Single-coordinate detailed balance.** Let $x$ and $x'$ differ only at coordinate $i$, where $x_i' = y$ for some token $y$. The proposal probability is

$$q_i(y \mid x) = \frac{p_t^i(y \mid x_t) g(r_i(y; x_t))}{\sum_{z \in \text{candidates}} p_t^i(z \mid x_t) g(r_i(z; x_t))},$$

where $r_i(y; x_t) = \frac{W_{\eta_t, \omega}(x_t^{(i \leftarrow y)})}{W_{\eta_t, \omega}(x_t)}$ and $g$ satisfies $g(u) = u \cdot g(1/u)$.

The acceptance probability is

$$\alpha_i(x, x') = \min \left\{ 1, \frac{\pi_{\eta, \omega}(x') q_i(x_i \mid x')}{\pi_{\eta, \omega}(x) q_i(y \mid x)} \right\}.$$

By the symmetry property of $g$ and the construction of the proposal, we have

$$\frac{q_i(y \mid x)}{q_i(x_i \mid x')} = \frac{W_{\eta, \omega}(x')}{W_{\eta, \omega}(x)}.$$

Since $\pi_{\eta, \omega}(x) = Z^{-1} p_1(x) W_{\eta, \omega}(x)$, it follows that

$$\frac{\pi_{\eta, \omega}(x') q_i(x_i \mid x')}{\pi_{\eta, \omega}(x) q_i(y \mid x)} = 1.$$

Therefore, $\alpha_i(x, x') = 1$ and detailed balance is satisfied.

**Step 2: Random-scan mixture.** The overall kernel is $K(x, x') = \frac{1}{L} \sum_{i=1}^{L} K_i(x, x')$, where $K_i$ is the kernel for updating coordinate $i$. Since each $K_i$ satisfies detailed balance with respect to $\pi_{\eta, \omega}$, their convex combination also satisfies detailed balance and hence preserves invariance. □

**Theorem (Convergence to Pareto Front).** Fix any $\omega \in \text{int} \, \Delta^{N-1}$ with strictly positive entries and let $S_\omega(x) = \min_n \omega_n \tilde{s}_n(x)$. If $\eta \to \infty$, samples drawn from $\pi_{\eta, \omega}(x) \propto p_1(x) \exp(\eta S_\omega(x))$ concentrate on the set

$$\mathcal{F}_\omega = \arg \max_x S_\omega(x),$$

and every element of $\mathcal{F}_\omega$ is Pareto optimal.

*Proof.* **Step 1: Maximizers of $S_\omega$ are Pareto optimal.** Suppose $x^* \in \mathcal{F}_\omega$ but $x^*$ is not Pareto optimal. Then there exists $y \in \mathcal{S}$ with

$$\tilde{s}_n(y) \geq \tilde{s}_n(x^*) \, \forall n, \quad \text{and} \quad \tilde{s}_m(y) > \tilde{s}_m(x^*) \text{ for some } m.$$

Since $\omega_n > 0$ for all $n$, multiplying preserves inequalities. If $m$ is the bottleneck coordinate of $x^*$, then $S_\omega(y) > S_\omega(x^*)$, contradiction. Otherwise, equality requires special weight alignments (measure zero). Thus maximizers are Pareto optimal almost surely.

**Step 2: Concentration as $\eta \to \infty$.** Let $S_\omega^* = \max_x S_\omega(x)$ and $\Delta_\omega = S_\omega^* - \max_{x \notin \mathcal{F}_\omega} S_\omega(x) > 0$. Then for $x \notin \mathcal{F}_\omega$,

$$\pi_{\eta, \omega}(x) \leq e^{-\eta \Delta_\omega} \cdot \frac{p_1(x)}{\sum_{z \in \mathcal{F}_\omega} p_1(z)}.$$

Summing gives $\pi_{\eta, \omega}(\mathcal{S} \setminus \mathcal{F}_\omega) \to 0$ as $\eta \to \infty$. Hence the mass concentrates on $\mathcal{F}_\omega$. □

**Theorem (Pareto Point Representability).** For every Pareto-optimal state $x^\dagger \in \mathcal{P}$ there exists $\omega \in \Delta^{N-1}$ such that $x^\dagger \in \arg \max_x S_\omega(x)$. Moreover, if $\tilde{s}_n(x^\dagger) > 0$ for all $n$, then $x^\dagger$ can be made the unique maximizer.

*Proof.* If $\tilde{s}_n(x^\dagger) > 0$, define

$$\omega_n = \frac{1/\tilde{s}_n(x^\dagger)}{\sum_{k=1}^N 1/\tilde{s}_k(x^\dagger)}.$$

Then $S_\omega(x^\dagger) = \frac{1}{\sum_k 1/\tilde{s}_k(x^\dagger)}$, and for any $y \neq x^\dagger$, some $m$ satisfies $\tilde{s}_m(y) < \tilde{s}_m(x^\dagger)$, implying $S_\omega(y) < S_\omega(x^\dagger)$. If some $\tilde{s}_n(x^\dagger) = 0$, perturb objectives by $\varepsilon > 0$ and take the limit. □

**Theorem (Coverage Guarantee).** Let $\mu$ be any probability distribution with full support on $\text{int} \, \Delta^{N-1}$. If $\omega \sim \mu$ and $\eta \to \infty$, then the induced sampler visits every Pareto-optimal state with positive probability.

*Proof.* By representability, each Pareto point $x^\dagger$ maximizes $S_\omega$ for some interior $\omega^\dagger$. By continuity, there exists a neighborhood $U_{x^\dagger}$ where $x^\dagger$ remains optimal. Since $\mu(U_{x^\dagger}) > 0$, randomizing $\omega$ ensures $x^\dagger$ is visited with positive probability in the high-$\eta$ limit. $\square$

**Remark.** The guarantees hold for any finite $\mathcal{S}$ and bounded objectives. In practice, convergence depends on the chain mixing rate, the annealing schedule for $\eta$, and the choice of balancing function $g$.

## C  PepReDi and SMILESReDi Generate Diverse and Biologically Plausible Sequences

To enable the efficient generation of peptide binders, we developed an unconditional peptide generator, **PepReDi**, based on the ReDi framework. The model backbone of PepReDi is a Diffusion Transformer (DiT) architecture (Peebles & Xie, 2022). We trained PepDFM on a custom dataset comprising approximately 15,000 peptides from the PepNN and BioLip2 datasets, as well as sequences from the PPIRef dataset, with lengths ranging from 6 to 49 amino acids (Abdin et al., 2022; Zhang et al., 2024; Bushuiev et al., 2023). Using this trained model, we generated new data couplings containing 10,000 sequences for each peptide length and used them to fine-tune PepReDi in an iterative rectification procedure. This rectification was performed three times and yielded substantial improvements in training loss, validation negative log-likelihood (NLL), perplexity (PPL), and conditional TC (Table S2). Notably, the conditional TC rises after the first rectification, likely due to the distributional shift from the large, model-generated coupling, whose absolute TC can be higher even though ReDi guarantees a monotonic decrease within each coupling. The low validation NLL and PPL metrics showcase PepReDi's reliability to generate biologically plausible wild-type peptide sequences.

SMILESReDi adopts the same backbone structure as PepReDi, enhanced with Rotary Positional Embeddings (RoPE), which effectively captures the relative inter-token interactions in peptide SMILES (Su et al., 2024). SMILESReDi also incorporates a time-dependent noising schedule to improve its capability to generate valid peptide SMILES sequences (D.2). We applied the same training data as PepMDLM, a state-of-the-art diffusion model that generates valid peptide SMILES sequences (Tang et al., 2025b). After only two training epochs, SMILESReDi converged to a validation NLL of 0.722 and achieved a sampling validity of 76.3% using just 16 generation steps. One hundred SMILES sequences were then generated by the trained SMILESReDi for each length from 4 to 1035, forming a large and diverse new data coupling. Following a single round of rectification, the validation NLL further decreased to 0.608, and the sampling validity rose dramatically to 98.6% with 16 steps and 100% with 32 steps S3. While its similarity-to-nearest-neighbor (SNN) score and diversity are comparable to those of PepMDLM (details on metrics are provided in Appendix D.2), SMILESReDi substantially outperforms PepMDLM in validity, highlighting its superior capability of generating diverse chemically-modified peptide SMILES sequences.

## D  Base Model Details

### D.1  PepReDi

**Model Architecture.** The backbone of PepReDi is built on a Diffusion Transformer (DiT) framework implemented within a Masked Diffusion Language Model (MDLM) paradigm (Peebles & Xie, 2022; Sahoo et al., 2024). Input amino acid sequences are transformed to discrete tokens using the ESM-2-650M tokenizer (Lin et al., 2023). Tokenized amino acid sequences and time-steps are converted to continuous embedding vectors using two separate layers, which are then fused and processed by stacked DiT transformer blocks equipped with multi-head self-attention to capture long-range dependencies in the amino-acid sequence. Residual connections and layer normalization stabilize the training dynamics, and a final projection layer outputs token logits for each position.

**Dataset Curation.** The dataset for PepReDi training was curated from the PepNN, BioLip2, and PPIRef dataset (Abdin et al., 2022; Zhang et al., 2024; Bushuiev et al., 2023). All peptides from PepNN and BioLip2 were included, along with sequences from PPIRef ranging from 6 to 49 amino acids in length. The dataset was divided into training, validation, and test sets at an 80/10/10 ratio.

**Training Strategy.** Training was conducted on a single node equipped with one NVIDIA GPU and 128 GB of GPU memory using the SLURM workload manager. The model was trained for 100 epochs using the Adam optimizer and a learning rate of 1e-4 with weight decay of 1e-5. A learning rate scheduler with 10 warm-up epochs and cosine decay was used, with initial and minimum learning rates both 1e-5. The network architecture included a model dimension of 512, 6 transformer layers, and 8 attention heads, with a vocabulary size of 24 and a maximum sequence length of 100 tokens. Conditional total correlation estimation was performed using 20 batches and 50 samples per batch to monitor rectification quality during training. The model checkpoint with the lowest total correlation was saved. For training rectified models, the same hyperparameter setting was applied, except for the loaded pre-trained model checkpoint and the weight decay being increased to 2e-5.

**Dynamic Batching.** To enhance computational efficiency and manage variable-length token sequences, we implemented dynamic batching. Drawing inspiration from ESM-2's approach (Lin et al., 2023), input peptide sequences were sorted by length to optimize GPU memory utilization, with a maximum token size of 100 per GPU.

**Rectification.** The trained model applied 16 sampling steps to generate 10k sequences for each peptide length, ranging from 6 to 49, with a temperature hyperparameter set to 1. After generation, dynamic batching was used to optimize GPU memory utilization for future rectified training.

### D.2 SMILESReDi

**Model Architecture.** SMILESReDi follows the ReDi paradigm and uses a Diffusion Transformer (DiT) backbone embedded in a Masked Diffusion Language Model (MDLM) design to generate molecular SMILES sequences (Peebles & Xie, 2022; Sahoo et al., 2024). Input SMILES sequences are transformed to discrete tokens using the PeptideCLM -23M tokenizer. Tokenized amino acid sequences and time-steps are converted to continuous embedding vectors using two separate layers. Both embeddings are then fused and processed by stacked DiT transformer blocks that incorporate Rotary Positional Embeddings (RoPE) and multi-head attention modules to capture long-range structural dependencies while preserving positional information (Su et al., 2024). A final layer normalization and linear projection outputs token logits for each position.

**Time-dependent bond-aware noising schedule.** Peptide SMILES share a conserved backbone of alternating carbonyl and amide groups connected by chemically constrained peptide bonds, while their side chains remain highly diverse. Standard discrete flow matching can corrupt these critical bond tokens too early, hindering the flow from recovering the backbone along the probability path. Inspired by previous work in bond-dependent masking, we devised a time-dependent bond-aware noising schedule that preserves backbone tokens longer than side-chain tokens, allowing the model to reconstruct the invariant scaffold before generating variable side chains. Specifically, for each position $j$ with a bond indicator $b_j \in \{0, 1\}$, the time-$t$ marginal of the probability path is

$$p_t(x_t^{(j)} \mid x_0^{(j)}, x_1^{(j)}) = \left[ b_j t^\gamma + (1 - b_j)t \right] \delta_{x_1^{(j)}} + \left[ 1 - b_j t^\gamma - (1 - b_j)t \right] \delta_{x_0^{(j)}}, \quad t \in [0, 1], \; \gamma > 1,$$

so each token is equal to $x_1^{(j)}$ with the indicated mixture coefficient and to $x_0^{(j)}$ otherwise, ensuring that backbone tokens ($b_j = 1$) transition more slowly than non-bond tokens along the DFM probability path.

**Training Strategy.** The training is conducted on a 4*A6000 NVIDIA RTX 6000 Ada GPU system with 48 GB of VRAM for 5 epochs. The model checkpoint with the lowest evaluation loss was saved. The Adam optimizer was employed with a learning rate of 1e-4. A learning rate scheduler with 10% total training steps and cosine decay was used, with initial and minimum learning rates both 1e-5. The network architecture included a model dimension of 768, 8 transformer layers, and 8 attention heads. Gradient clip value was set to 1.0 and $\gamma$ to 2.0 in the time-dependent bond-aware noising schedule. For training rectified models, the same hyperparameter setting was applied, except for the loaded pre-trained model checkpoint and the total training epochs set to 10.

**Rectification.** The trained model applied 100 sampling steps to generate 100 sequences for each peptide length, ranging from 4 to 1035, with a temperature hyperparameter set to 1. After generation, dynamic batching was used to optimize GPU memory utilization for future rectified training.

**Evaluation Metrics.**

- **Validity** is defined as the fraction of peptide SMILES that pass the SMILES2PEPTIDE filter (Tang et al., 2025b), indicating that it translates to a synthesizable peptide.

- **Uniqueness** is defined as the fraction of mutually distinct peptide SMILES.

- **Diversity** is defined as one minus the average Tanimoto similarity between the Morgan fingerprints of every pair of generated sequences, which measures the similarity in structure across generated peptides.

$$\text{Diversity} = 1 - \frac{1}{\binom{N_{\text{generated}}}{2}} \sum_{i,j} \frac{\mathbf{f}(\mathbf{x}_i) \cdot \mathbf{f}(\mathbf{x}_j)}{|\mathbf{f}(\mathbf{x}_i)| + |\mathbf{f}(\mathbf{x}_j)| - \mathbf{f}(\mathbf{x}_i) \cdot \mathbf{f}(\mathbf{x}_j)}$$

  where $\mathbf{f}(\mathbf{x}_i)$ and $\mathbf{f}(\mathbf{x}_j)$ are the 2048-dimensional Morgan fingerprint with radius 3 for a pair of generated sequences $\mathbf{x}_i$ and $\mathbf{x}_j$.

- **Similarity to Nearest Neighbor (SNN)** is defined as the maximum Tanimoto similarity between a generated sequence $\mathbf{x}_i$ with a sequence in the dataset $\tilde{\mathbf{x}}_j$.

$$\text{SNN} = \max_{j \in |\mathcal{D}|} \left( \frac{\mathbf{f}(\mathbf{x}_i) \cdot \mathbf{f}(\tilde{\mathbf{x}}_j)}{|\mathbf{f}(\mathbf{x}_i)| + |\mathbf{f}(\tilde{\mathbf{x}}_j)| - \mathbf{f}(\mathbf{x}_i) \cdot \mathbf{f}(\tilde{\mathbf{x}}_j)} \right)$$

### D.3 PEPDFM

**Model Architecture.** The base model is a time-dependent architecture based on U-Net (Ronneberger et al., 2015). It uses two separate embedding layers for sequence and time, followed by five convolutional blocks with varying dilation rates to capture temporal dependencies, while incorporating time-conditioning through dense layers. The final output layer generates logits for each token. We used a polynomial convex schedule with a polynomial exponent of 2.0 for the mixture discrete probability path in the discrete flow matching.

**Dataset Curation.** The dataset for PepDFM training was curated from the PepNN, BioLip2, and PPIRef dataset (Abdin et al., 2022; Zhang et al., 2024; Bushuiev et al., 2023). All peptides from PepNN and BioLip2 were included, along with sequences from PPIRef ranging from 6 to 49 amino acids in length. The dataset was divided into training, validation, and test sets at an 80/10/10 ratio.

**Training Strategy.** The training is conducted on a 2xH100 NVIDIA NVL GPU system with 94 GB of VRAM for 200 epochs with batch size 512. The model checkpoint with the lowest evaluation loss was saved. The Adam optimizer was employed with a learning rate 1e-4. A learning rate scheduler with 20 warm-up epochs and cosine decay was used, with initial and minimum learning rates both 1e-5. The embedding dimension and hidden dimension were set to be 512 and 256 respectively for the base model.

**Performance.** PepDFM achieved a validation loss of 3.1051. Its low generalized KL loss during evaluation demonstrates PepDFM's strong capability to generate sequences with high biological plausibility (Gat et al., 2024).

## E    OBJECTIVE DESCRIPTION

In this work, five key property objectives are considered in the peptide binder tasks: hemolysis, non-fouling, solubility, half-life, and binding affinity. Each of these properties plays a crucial role in optimizing the therapeutic potential of peptides. Hemolysis refers to the peptide's ability to minimize red blood cell lysis, ensuring safe systemic circulation (Pirtskhalava et al., 2013). Non-fouling properties describe the peptide's resistance to unwanted interactions with biomolecules, thus enhancing its stability and bioavailability in vivo (Chen et al., 2009). Solubility is critical for ensuring adequate peptide dissolution in biological fluids, directly influencing its absorption and therapeutic efficacy (Fosgerau & Hoffmann, 2015). Half-life indicates the duration for which the peptide remains

active in circulation, which is vital for reducing dosing frequency (Swanson, 2014). Finally, binding affinity measures the strength of the peptide's interaction with its target, directly correlating to its biological activity and potency in therapeutic applications (Bostrom et al., 2008).

# F   SCORE MODEL DETAILS

We applied the score models from Tang et al. (2025b) to guide the generation of chemically-modified peptide binders. We now introduce the score model developed for the wild-type peptide binder generation task. We collected hemolysis (9,316), non-fouling (17,185), solubility (18,453), and binding affinity (1,781) data for classifier training from the PepLand and PeptideBERT datasets (Zhang et al., 2023; Guntuboina et al., 2023). All sequences taken are wild-type L-amino acids and are tokenized and represented by the ESM-2 protein language model (Lin et al., 2023).

## F.1   BOOSTED TREES FOR CLASSIFICATION

For hemolysis, non-fouling, and solubility classification, we trained XGBoost boosted tree models for logistic regression. We split the data into 0.8/0.2 train/validation using stratified splits from scikit-learn (Pedregosa et al., 2011) and generated mean-pooled ESM-2-650M (Lin et al., 2023) embeddings as input features to the model. We ran 50 trials of OPTUNA (Akiba et al., 2019) search to determine the optimal XGBoost hyperparameters (Table S1), tracking the best binary classification F1 scores. The best models for each property reached F1 scores of 0.58, 0.71, and 0.68 on the validation sets respectively.

Table S1: XGBoost Hyperparameters for Classification

| Hyperparameter | Value/Range |
| --- | --- |
| Objective | `binary:logistic` |
| Lambda | $[1e{-}8, 10.0]$ |
| Alpha | $[1e{-}8, 10.0]$ |
| Colsample by Tree | $[0.1, 1.0]$ |
| Subsample | $[0.1, 1.0]$ |
| Learning Rate | $[0.01, 0.3]$ |
| Max Depth | $[2, 30]$ |
| Min Child Weight | $[1, 20]$ |
| Tree Method | `hist` |

## F.2   BINDING AFFINITY SCORE MODEL

We developed an unpooled reciprocal attention transformer model to predict protein-peptide binding affinity, leveraging latent representations from the ESM-2 650M protein language model (Lin et al., 2023). Instead of relying on pooled representations, the model retains unpooled token-level embeddings from ESM-2, which are passed through convolutional layers followed by cross-attention layers. The binding affinity data were split into a 0.8/0.2 ratio, maintaining similar affinity score distributions across splits. We used OPTUNA (Akiba et al., 2019) for hyperparameter optimization, tracing validation correlation scores. The final model was trained for 50 epochs with a learning rate of 3.84e-5, a dropout rate of 0.15, 3 initial CNN kernel layers (dimension 384), 4 cross-attention layers (dimension 2048), and a shared prediction head (dimension 1024) in the end. The classifier reached 0.64 Spearman's correlation score on validation data.

## F.3   HALF-LIFE SCORE MODEL

**Dataset Curation.** The half-life dataset is curated from three publicly available datasets: PEPLife, PepTherDia, and THPdb2 (Mathur et al., 2016; D'Aloisio et al., 2021; Jain et al., 2024). Data related

to human subjects were selected, and entries with missing half-life values were excluded. After removing duplicates, the final dataset consists of 105 entries.

**Pre-training on stability data.** Given the small size of the half-life dataset, which is insufficient for training a model to capture the underlying data distribution, we first pre-trained a score model on a larger stability dataset to predict peptide stability (Tsuboyama et al., 2023). The model consists of three linear layers with ReLU activation functions, and a dropout rate of 0.3 was applied. The model was trained on a 2xH100 NVIDIA NVL GPU system with 94 GB of VRAM for 50 epochs. The Adam optimizer was employed with a learning rate of 1e-2. A learning rate scheduler with 5 warm-up epochs and cosine decay was used, with initial and minimum learning rates both 1e-3. After training, the model achieved a validation Spearman's correlation of 0.7915 and an $R^2$ value of 0.6864, demonstrating the reliability of the stability score model.

**Fine-tuning on half-life data.** The pre-trained stability score model was subsequently fine-tuned on the half-life dataset. Since half-life values span a wide range, the model was adapted to predict the base-10 logarithm of the half-life (h) values to stabilize the learning process. After fine-tuning, the model achieved a validation Spearman's correlation of 0.8581 and an $R^2$ value of 0.5977.

## G  Sampling Details

**Score Model Settings.** We cap the predicted log-scale half-life at 2 (i.e., 100 h) to prevent it from dominating the optimization and ensure balanced trade-offs across all properties. For the remaining objectives, hemolysis, non-fouling, solubility, and binding affinity, we directly employ their model outputs during sampling.

**Wild-Type Peptide Binder Generation Task Settings.** The total sampling steps are set to 20 multiplied by the binder length. All possible candidate token transitions are evaluated during each sampling step. We applied the same weight for each objective in all wild-type peptide binder generation tasks.

**Chemically-Modified Peptide Binder Generation Task Settings.** The total sampling steps are set to 128. With a vocabulary size of 586, evaluating all the possible candidate tokens is too computationally intensive. We therefore only evaluated the top 200 candidate tokens during each sampling step. We applied weight 0.7 for binding affinity, and 0.1 for hemolysis, non-fouling, and solubility, respectively. Instead of random initialization, the initial sequences $x_0$ are sampled from the pre-trained SMILESReDi[1] with 16 generation steps. During generation, AReUReDi rejects any transitions that will make the SMILES sequence an invalid peptide.

## H  Ablation Studies

**Effect of Rectification.**  To determine whether rectification improves over standard discrete flow matching, we compare AReUReDi using three generators: the base PepReDi model (no rectification), PepReDi after three rounds of rectification, and PepDFM, a standard discrete flow matching model trained on the same data following Gat et al. (2024) (Appendix D.3). We design wild-type binders for two distinct protein targets, 5AZ8 and AMHR2 (Table S7). For AMHR2, the rectified model achieves the best scores across all five properties, and improves predicted half-life by nearly 13 hours over the next-best method. For 5AZ8, rectification yields a substantially higher half-life while maintaining comparable performance on the remaining metrics. These results suggest that by lowering conditional total correlation and improving the probability path quality, rectification enables stronger Pareto trade-offs on the most demanding objectives.

**Annealed vs. Fixed Guidance Strength.**  We next ablate the guidance strength schedule by comparing the default annealed $\eta_t$ against fixed guidance strengths set to $\eta_{\min}$, $\eta_{\max}$, and their midpoint (Table S8). We evaluate on two settings: a structured protein with known binders (PDB 1DDV) and an intrinsically disordered protein without known binders (P53). Across both targets, no fixed-$\eta$ setting matches the annealed schedule. For 1DDV, annealing yields markedly higher half-life and the best solubility, while maintaining hemolysis, non-fouling, and affinity that meet or exceed all fixed-$\eta$ settings. The same trend holds for P53, where annealing consistently delivers the strongest performance across objectives. Overall, gradually increasing guidance strength improves Pareto

trade-offs, boosting challenging properties such as half-life without sacrificing other therapeutic metrics.

**Computational Budget.** To characterize the quality-compute trade-off, we vary the number of sampling steps on a wild-type binder design task (MYC, 12-mers) and a chemically-modified binder design task (NCAM1, length 200) (Table S10). Increasing the step budget improves the optimized properties in both settings, while runtime scales approximately linearly. For the more expensive SMILES task, improvements from 128 to 256 steps are marginal relative to the added compute, motivating our default choice of 128-256 steps for wild-type binders and 128 steps for chemically-modified binders in the main experiments (Appendix G).

**Sensitivity to Weight Vectors.** Finally, we verify that changing the Tchebycheff weight vector $\omega$ steers AReUReDi to different regions of the Pareto front by varying $\omega$ on two three-objective tasks: wild-type peptide binder design for CLK1 and chemically-modified binder design for GFAP (Tables S11, S12). Balanced weights yield balanced improvements, while emphasizing a single objective systematically shifts designs toward that objective with corresponding trade-offs in the others. This confirms that $\omega$ provides controllable navigation of Pareto trade-offs rather than merely re-sampling a fixed compromise.

**Role of the ReDi Prior.** We ablate the ReDi prior by replacing the learned prior $p_1(x)$ with an uniform prior while keeping the rest of AReUReDi unchanged, across wild-type binder (PPP5, 1B8Q) and chemically-modified binder (TfR, GLP1) design tasks (Table S13, S14). Using the learned prior consistently improves multi-objective outcomes, indicating that the flow prior is not redundant in the reward-tilted sampler but instead anchors generation in realistic, high-quality regions of sequence space.

Table S2: Training and validation performance of PepReDi over successive rectification rounds. Each row reports the training loss, validation negative log-likelihood (NLL), validation perplexity (PPL), and conditional total correlation (TC). PepReDi without superscript denotes the base model, while PepReDi[1], PepReDi[3], PepReDi[3] indicate the first, second, and third rounds of rectification, respectively.

|  | Train Loss | Val NLL | Val PPL | Conditional TC |
|---|---|---|---|---|
| PepReDi | 1.6567 | 1.6458 | 5.19 | 10.6027 |
| PepReDi[1] | 1.6170 | 1.6101 | 5.00 | 12.6250 |
| PepReDi[2] | 1.5347 | 1.5238 | 4.59 | 11.7279 |
| PepReDi[3] | **1.3538** | **1.3548** | **3.88** | **11.2339** |

Table S3: Evaluation metrics for the generative quality of peptide SMILES sequences of max token length set to 200. SMILESReDi without superscription denotes the base model, while SMILESReDi[1] refers to the model that has undergone one round of rectification.

| Model | Validity ($\uparrow$) | Uniqueness ($\uparrow$) | Diversity ($\uparrow$) | SNN ($\downarrow$) |
|---|---|---|---|---|
| Data | 1.000 | 1.000 | 0.885 | 1.000 |
| PepMDLM | 0.450 | 1.000 | 0.705 | 0.513 |
| **SMILESReDi** | **0.763** | 1.000 | 0.719 | 0.593 |
| **SMILESReDi**[1] | **0.986** | 1.000 | 0.665 | 0.579 |
| PepTune | 1.000 | 1.000 | 0.677 | 0.486 |
| **AReUReDi** | 1.000 | 1.000 | **0.789** | **0.392** |

Table S4: **Adding a sampling constraint greatly improves AReUReDi's performance.** Wild-type binders for two protein targets (PDB 8CN1 and 4EBP2) were generated with or without a sampling constraint using the same number of generation steps. The table reports the average score for each objective, calculated from 100 generated binders per setting. The best score for each objective is highlighted in bold.

| Target | Method | Hemolysis | Non-Fouling | Solubility | Half-Life (h) | Affinity |
|---|---|---|---|---|---|---|
| 8CN1 | w/o constraints | 0.8650 | 0.4782 | 0.4627 | 2.54 | 5.2412 |
|  | w/ constraints | **0.9213** | **0.8676** | **0.8697** | **44.70** | **5.5143** |
| 4EBP2 | w/o constraints | 0.8879 | 0.4288 | 0.4257 | 1.8781 | 5.7132 |
|  | w/ constraints | **0.9356** | **0.8767** | **0.8692** | **53.95** | **6.4571** |

Table S5: Ablation results for wild-type peptide binder design targeting PDB 7LUL with different guidance settings. For each setting, 100 binders of length 7 were designed.

| Guidance Settings | | | Hemolysis | Solubility | Affinity |
| Hemolysis | Solubility | Affinity | | | |
|:---:|:---:|:---:|:---:|:---:|:---:|
| ✓ | ✓ | ✓ | 0.9389 | 0.9398 | 6.2559 |
| × | ✓ | ✓ | 0.8964 | 0.9465 | 6.3272 |
| ✓ | × | ✓ | 0.9502 | 0.4013 | 6.9798 |
| ✓ | ✓ | × | 0.9535 | 0.9642 | 5.2611 |
| × | × | ✓ | 0.8812 | 0.2877 | 7.5057 |
| × | ✓ | × | 0.9036 | 0.9725 | 5.2449 |
| ✓ | × | × | 0.9802 | 0.6135 | 5.0985 |
| × | × | × | 0.8431 | 0.5810 | 4.8919 |

Table S6: Ablation results for wild-type peptide binder design targeting PDB CLK1 with different guidance settings. For each setting, 100 binders of length 12 were designed.

| Guidance Settings | | | Non-Fouling | Half-Life (h) | Affinity |
| Non-Fouling | Half-Life (h) | Affinity | | | |
|:---:|:---:|:---:|:---:|:---:|:---:|
| ✓ | ✓ | ✓ | 0.8285 | 74.04 | 6.8099 |
| × | ✓ | ✓ | 0.2902 | 96.59 | 7.3906 |
| ✓ | × | ✓ | 0.9365 | 1.33 | 7.2029 |
| ✓ | ✓ | × | 0.9479 | 75.68 | 6.3437 |
| × | × | ✓ | 0.9625 | 1.23 | 6.2319 |
| × | ✓ | × | 0.3540 | 100.00 | 6.4116 |
| ✓ | × | × | 0.2531 | 2.96 | 8.6580 |
| × | × | × | 0.4988 | 1.82 | 5.4739 |

Table S7: **Rectification of the base generation model improves AReUReDi's performance.** Wild-type binders for two protein targets (PDB 5AZ8 and AMHR2) were generated using AReUReDi with three different base models: PepDFM, PepReDi (without rectification), and PepReDi[3] (with three rounds of rectification). The table reports the average score for each objective, calculated from 100 generated binders per setting. The best score for each objective is highlighted in bold.

| Target | Base Model | Hemolysis | Non-Fouling | Solubility | Half-Life (h) | Affinity |
|---|---|---|---|---|---|---|
| | PepDFM | 0.9296 | **0.8867** | **0.8743** | 37.30 | 6.2291 |
| 5AZ8 | PepReDi | **0.9326** | 0.8759 | 0.8572 | 50.16 | **6.4391** |
| | PepReDi[3] | 0.9293 | 0.8732 | 0.8605 | **58.33** | 6.2792 |
| | PepDFM | 0.9412 | 0.8774 | 0.8612 | 47.84 | 7.2373 |
| AMHR2 | PepReDi | 0.9127 | 0.8602 | 0.8460 | 50.92 | 7.0101 |
| | PepReDi[3] | **0.9420** | **0.8914** | **0.8755** | **63.34** | **7.2533** |

Table S8: **Annealed guidance strength improves AReUReDi's performance.** Wild-type binders for two protein targets (PDB 1DDV and P53) were generated under four guidance schedules: (1) fixed at the minimum strength $\eta_{min} = 1.0$, (2) fixed at the maximum strength $\eta_{max} = 20.0$, (3) fixed at the midpoint $\frac{1}{2}(\eta_{min} + \eta_{max}) = 10.5$, and (4) an annealed schedule where $\eta_t$ increases from $\eta_{min}$ to $\eta_{max}$ over optimization steps. The table reports the average score for each objective, calculated from 100 generated binders per setting. The best score for each objective is highlighted in bold.

| Target | Method | Hemolysis | Non-Fouling | Solubility | Half-Life (h) | Affinity |
|---|---|---|---|---|---|---|
| | $\eta = \eta_{min}$ | 0.9130 | 0.8575 | 0.8429 | 38.70 | 5.3554 |
| 1DDV | $\eta = \eta_{max}$ | 0.9156 | 0.8512 | 0.8479 | 40.27 | 5.4359 |
| | $\eta = \frac{1}{2}(\eta_{min} + \eta_{max})$ | 0.9108 | 0.8641 | 0.8544 | 40.43 | 5.5396 |
| | $\eta_t = \eta_{\min} + (\eta_{\max} - \eta_{\min})\frac{t}{T-1}$ | 0.9128 | 0.8545 | **0.8565** | **44.73** | 5.4482 |
| | $\eta = \eta_{min}$ | 0.9335 | 0.8800 | 0.8706 | 49.97 | 6.2538 |
| P53 | $\eta = \eta_{max}$ | 0.9293 | 0.8693 | 0.8657 | 61.76 | 6.3043 |
| | $\eta = \frac{1}{2}(\eta_{min} + \eta_{max})$ | 0.9294 | 0.8713 | 0.8653 | 59.43 | 6.3060 |
| | $\eta_t = \eta_{\min} + (\eta_{\max} - \eta_{\min})\frac{t}{T-1}$ | **0.9353** | **0.8818** | **0.8785** | **62.83** | **6.3508** |

Table S9: **Best-of-$N$ comparison between PepTune+DPLM and AReUReDi under matched wall-clock time.** For each target, PepTune+DPLM is allowed to generate 100 binders while AReUReDi generates only 4 (PDB 1B8Q) or 3 (PPP5). Top-2 sequences from each method were reported. The table reports the average score for each objective.

| Target | Method | Rank | Hemolysis | Non-Fouling | Solubility | Half-Life (h) | Affinity |
|---|---|---|---|---|---|---|---|
| | PepTune + DPLM | Top 1 | 0.9323 | 0.4379 | 0.3624 | 9.82 | 7.0534 |
| | | Top 2 | 0.8718 | 0.2573 | 0.2391 | 38.67 | 6.5605 |
| 1B8Q | AReUReDi | Top 1 | 0.8651 | 0.8638 | 0.8892 | 100.00 | 5.6008 |
| | | Top 2 | 0.9354 | 0.8567 | 0.9331 | 49.25 | 6.5605 |
| | PepTune + DPLM | Top 1 | 0.7984 | 0.3338 | 0.2342 | 80.27 | 7.6117 |
| | | Top 2 | 0.7901 | 0.0966 | 0.1328 | 100.00 | 6.7571 |
| PPP5 | AReUReDi | Top 1 | 0.9407 | 0.9378 | 0.9131 | 100.00 | 6.8193 |
| | | Top 2 | 0.9606 | 0.8750 | 0.8399 | 90.16 | 6.8969 |

Table S10: **Increasing generation steps improves AReUReDi's performance.** AReUReDi designed 100 generated binders for MYC (12-mer wild-type peptides) and NCAM1 (chemically-modified peptides of length 200) using different numbers of generation steps. The table reports the average score for each objective. Half-life is not optimized for NCAM1 and is indicated by "*".

| Target | # Steps | Hemolysis | Non-Fouling | Solubility | Half-Life (h) | Affinity | Time |
|--------|---------|-----------|-------------|------------|---------------|----------|------|
| MYC | 64 | 0.9279 | 0.8571 | 0.8519 | 5.49 | 6.5167 | 67 |
| | 128 | 0.9301 | 0.8721 | 0.8627 | 16.54 | 6.5811 | 131 |
| | 256 | 0.9357 | 0.8820 | 0.8740 | 34.83 | 6.5293 | 265 |
| NCAM1 | 64 | 0.8801 | 0.2468 | 0.7954 | * | 5.3936 | 112 |
| | 128 | 0.8840 | 0.2657 | 0.8109 | * | 5.4377 | 198 |
| | 256 | 0.8900 | 0.3015 | 0.8202 | * | 5.5929 | 423 |

Table S11: Ablation results for wild-type peptide binder design targeting CLK1 with different weight vector settings. For each setting, 100 binders of length 12 were designed. The table reports the average score for each objective.

| Weight Vectors | | | Non-Fouling | Half-Life (h) | Affinity |
|----------------|-----------|----------|-------------|---------------|----------|
| Non-Fouling | Half-Life | Affinity | | | |
| 0.3 | 0.3 | 0.3 | 0.8285 | 74.04 | 6.8099 |
| 0.8 | 0.1 | 0.1 | 0.9367 | 6.94 | 6.5231 |
| 0.1 | 0.8 | 0.1 | 0.5642 | 85.47 | 6.3649 |
| 0.1 | 0.1 | 0.8 | 0.6698 | 48.94 | 7.4922 |

Table S12: Ablation results for chemically-modified peptide binder design targeting GFAP with different weight vector settings. For each setting, 100 binders of length 200 were designed. The table reports the average score for each objective.

| Weight Vectors | | | Non-Fouling | Solubility | Affinity |
|----------------|------------|----------|-------------|------------|----------|
| Non-Fouling | Solubility | Affinity | | | |
| 0.3 | 0.3 | 0.3 | 0.2754 | 0.8169 | 5.3011 |
| 0.8 | 0.1 | 0.1 | 0.3322 | 0.7528 | 5.3487 |
| 0.1 | 0.8 | 0.1 | 0.2273 | 0.8327 | 5.3378 |
| 0.1 | 0.1 | 0.8 | 0.2498 | 0.7910 | 5.8827 |

Table S13: **PepReDi provides prior knowledge that helps AReUReDi to generate samples with better multi-objective trade-offs.** 100 wild-type binders were designed for PDB 1B8Q (8-mer) and PPP5 (16-mer), respectively. The table reports the average score for each objective. The best score for each objective is highlighted in bold.

| Target | Prior | Hemolysis | Non-Fouling | Solubility | Half-Life (h) | Affinity |
|--------|-------|-----------|-------------|------------|---------------|----------|
| 1B8Q | Uniform Prior | 0.9009 | 0.8191 | 0.8049 | 14.20 | 5.8432 |
| | PepReDi Prior | **0.9214** | **0.8680** | **0.8654** | **22.93** | **5.7130** |
| PPP5 | Uniform Prior | 0.9265 | 0.8263 | 0.7993 | 17.52 | 6.7122 |
| | PepReDi Prior | **0.9412** | **0.896** | **0.8832** | **38.28** | **6.7186** |

Table S14: **SMILESReDi provides prior knowledge that helps AReUReDi to generate samples with better multi-objective trade-offs.** For each setting, 100 chemically-modified binders of length 200 were designed. The table reports the average score for each objective. The best score for each objective is highlighted in bold.

| Target | Prior | Hemolysis | Non-Fouling | Solubility | Affinity |
|---|---|---|---|---|---|
| TfR | Uniform Prior | 0.8652 | 0.2381 | **0.7777** | 5.5535 |
| | SMILESReDi Prior | **0.8665** | **0.3234** | 0.7408 | **6.1271** |
| GLP1 | Uniform Prior | 8.3414 | 0.2123 | **0.7777** | 7.5731 |
| | SMILESReDi Prior | **0.8743** | **0.3438** | 0.7661 | **8.3414** |

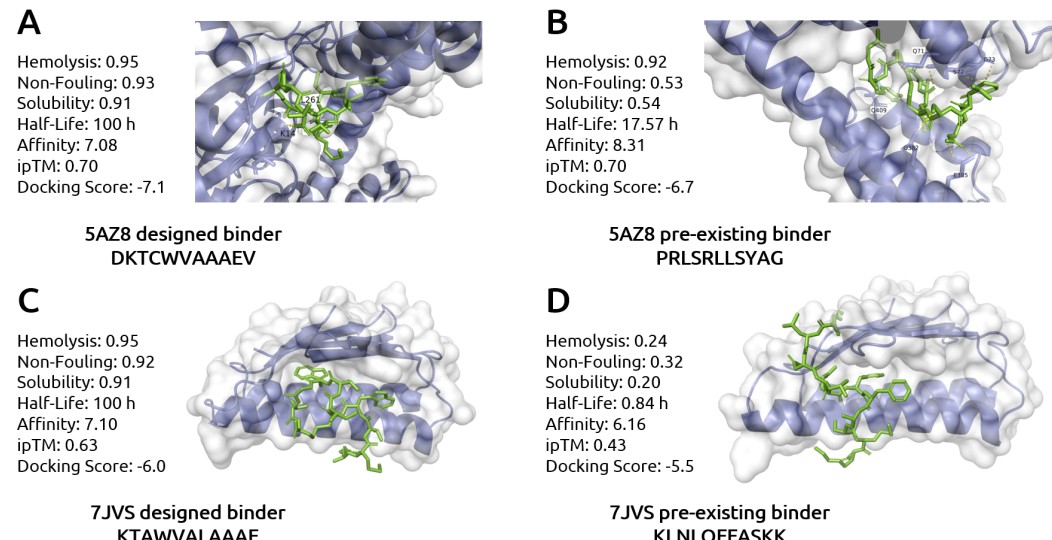

**A**

Hemolysis: 0.95
Non-Fouling: 0.93
Solubility: 0.91
Half-Life: 100 h
Affinity: 7.08
ipTM: 0.70
Docking Score: -7.1

5AZ8 designed binder
DKTCWVAAAEV

**B**

Hemolysis: 0.92
Non-Fouling: 0.53
Solubility: 0.54
Half-Life: 17.57 h
Affinity: 8.31
ipTM: 0.70
Docking Score: -6.7

5AZ8 pre-existing binder
PRLSRLLSYAG

**C**

Hemolysis: 0.95
Non-Fouling: 0.92
Solubility: 0.91
Half-Life: 100 h
Affinity: 7.10
ipTM: 0.63
Docking Score: -6.0

7JVS designed binder
KTAWVALAAAE

**D**

Hemolysis: 0.24
Non-Fouling: 0.32
Solubility: 0.20
Half-Life: 0.84 h
Affinity: 6.16
ipTM: 0.43
Docking Score: -5.5

7JVS pre-existing binder
KLNLQFFASKK

Figure S1: **Complex structures of target proteins with pre-existing binders. (A)-(B)** 5AZ8 **(C)-(D)** 7JVS. Each panel shows the complex structure of the target with either an AReUReDi-designed binder or its pre-existing binder. For each binder, five property scores are provided, as well as the ipTM score from AlphaFold3 and the docking score from AutoDock VINA. Interacting residues on the target are visualized.

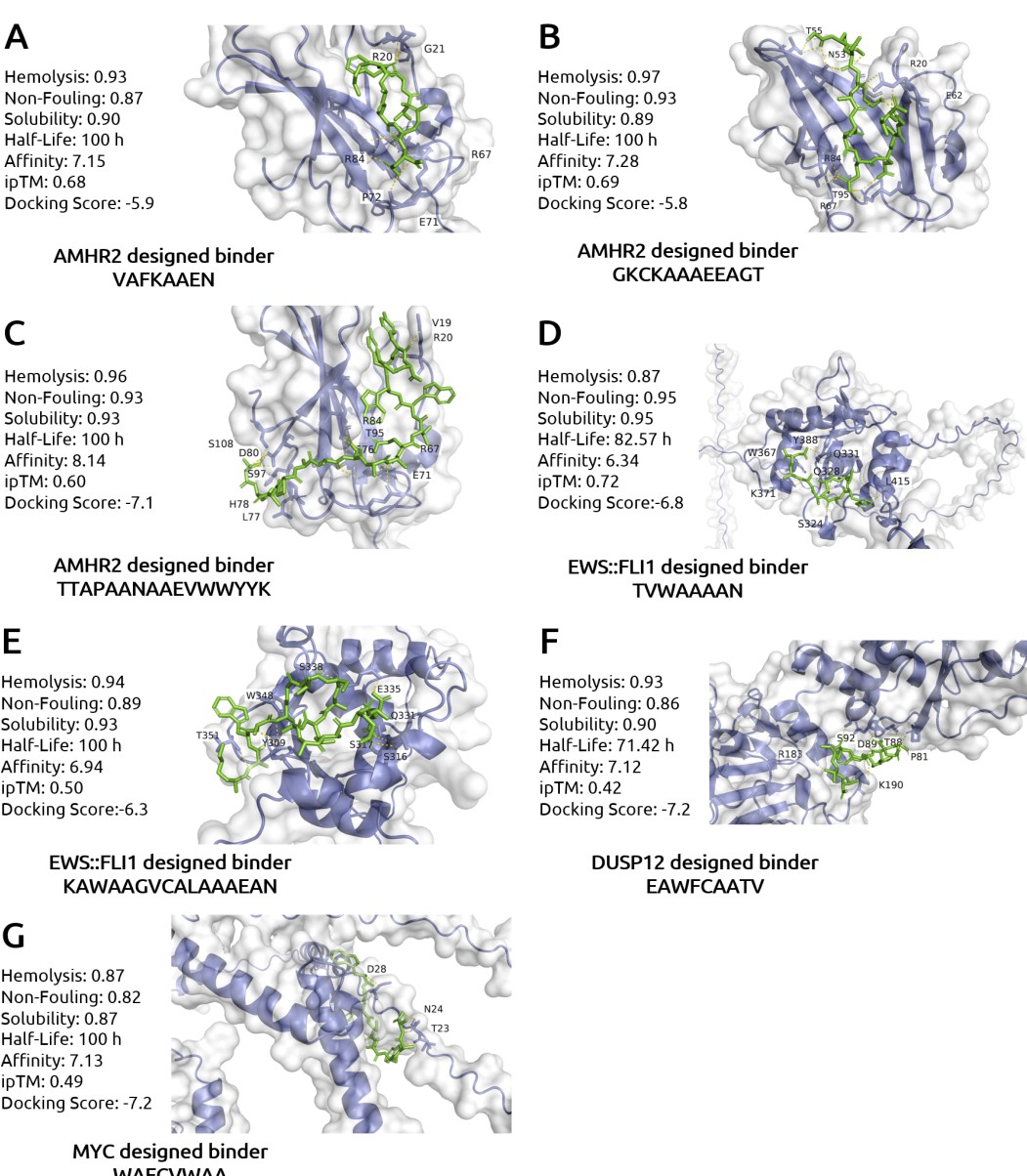

**A**

Hemolysis: 0.93
Non-Fouling: 0.87
Solubility: 0.90
Half-Life: 100 h
Affinity: 7.15
ipTM: 0.68
Docking Score: -5.9

**AMHR2 designed binder
VAFKAAEN**

**B**

Hemolysis: 0.97
Non-Fouling: 0.93
Solubility: 0.89
Half-Life: 100 h
Affinity: 7.28
ipTM: 0.69
Docking Score: -5.8

**AMHR2 designed binder
GKCKAAAEEAGT**

**C**

Hemolysis: 0.96
Non-Fouling: 0.93
Solubility: 0.93
Half-Life: 100 h
Affinity: 8.14
ipTM: 0.60
Docking Score: -7.1

**AMHR2 designed binder
TTAPAANAAEVWWYYK**

**D**

Hemolysis: 0.87
Non-Fouling: 0.95
Solubility: 0.95
Half-Life: 82.57 h
Affinity: 6.34
ipTM: 0.72
Docking Score:-6.8

**EWS::FLI1 designed binder
TVWAAAAN**

**E**

Hemolysis: 0.94
Non-Fouling: 0.89
Solubility: 0.93
Half-Life: 100 h
Affinity: 6.94
ipTM: 0.50
Docking Score:-6.3

**EWS::FLI1 designed binder
KAWAAGVCALAAAEAN**

**F**

Hemolysis: 0.93
Non-Fouling: 0.86
Solubility: 0.90
Half-Life: 71.42 h
Affinity: 7.12
ipTM: 0.42
Docking Score: -7.2

**DUSP12 designed binder
EAWFCAATV**

**G**

Hemolysis: 0.87
Non-Fouling: 0.82
Solubility: 0.87
Half-Life: 100 h
Affinity: 7.13
ipTM: 0.49
Docking Score: -7.2

**MYC designed binder
WAFCVWAA**

Figure S2: **Complex structures of target proteins without pre-existing binders. (A)-(C)** AMHR2, **(D)-(E)** EWS::FLI1, **(F)** MYC, **(G)** DUSP12. Each panel shows the complex structure of the target with an AReUReDi-designed binder. For each binder, five property scores are provided, as well as the ipTM score from AlphaFold3 and the docking score from AutoDock VINA. Interacting residues on the target are visualized.

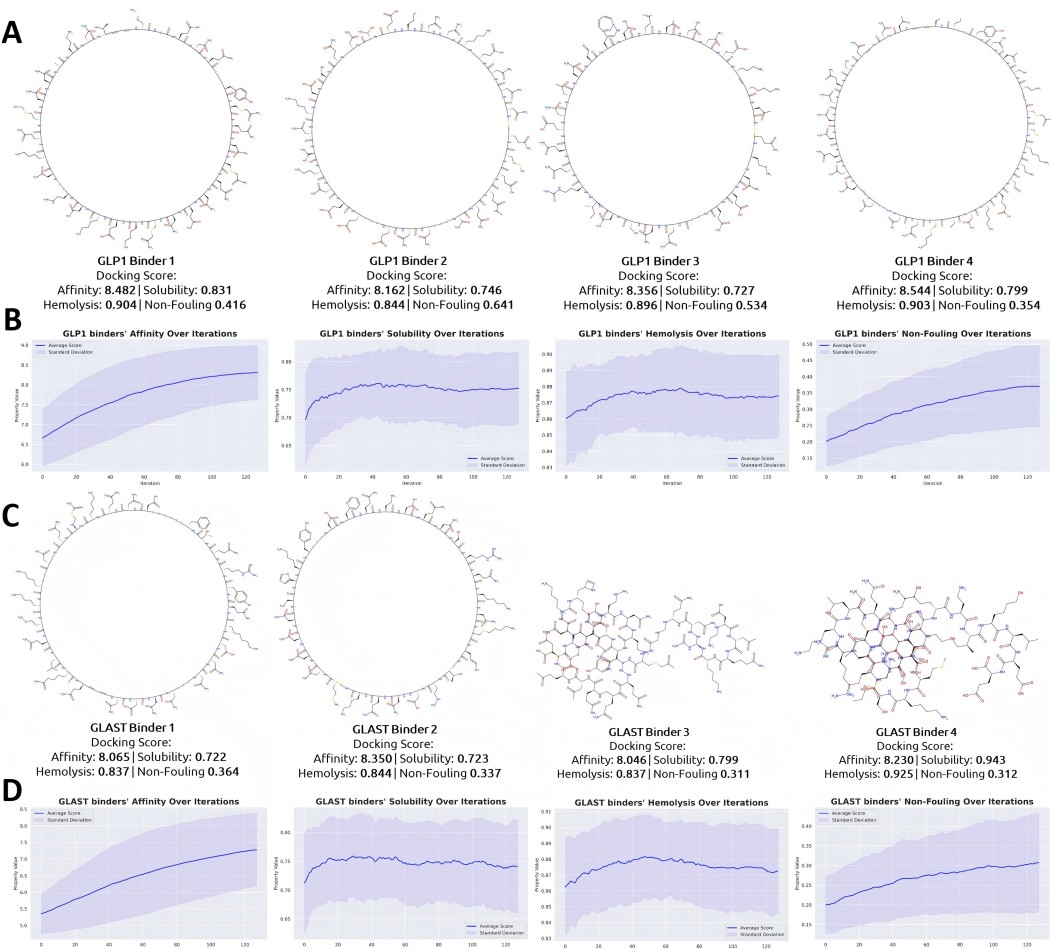

Figure S3: **(A), (C)** Example 2D SMILES structure of AReUReDi-designed peptide binders with four property scores for GLP1 and GLAST, respectively. **(B), (D)** Plots showing the mean scores for each property across the number of iterations during AReUReDi's design of binders of length 200 for GLP1 and GLAST, respectively.

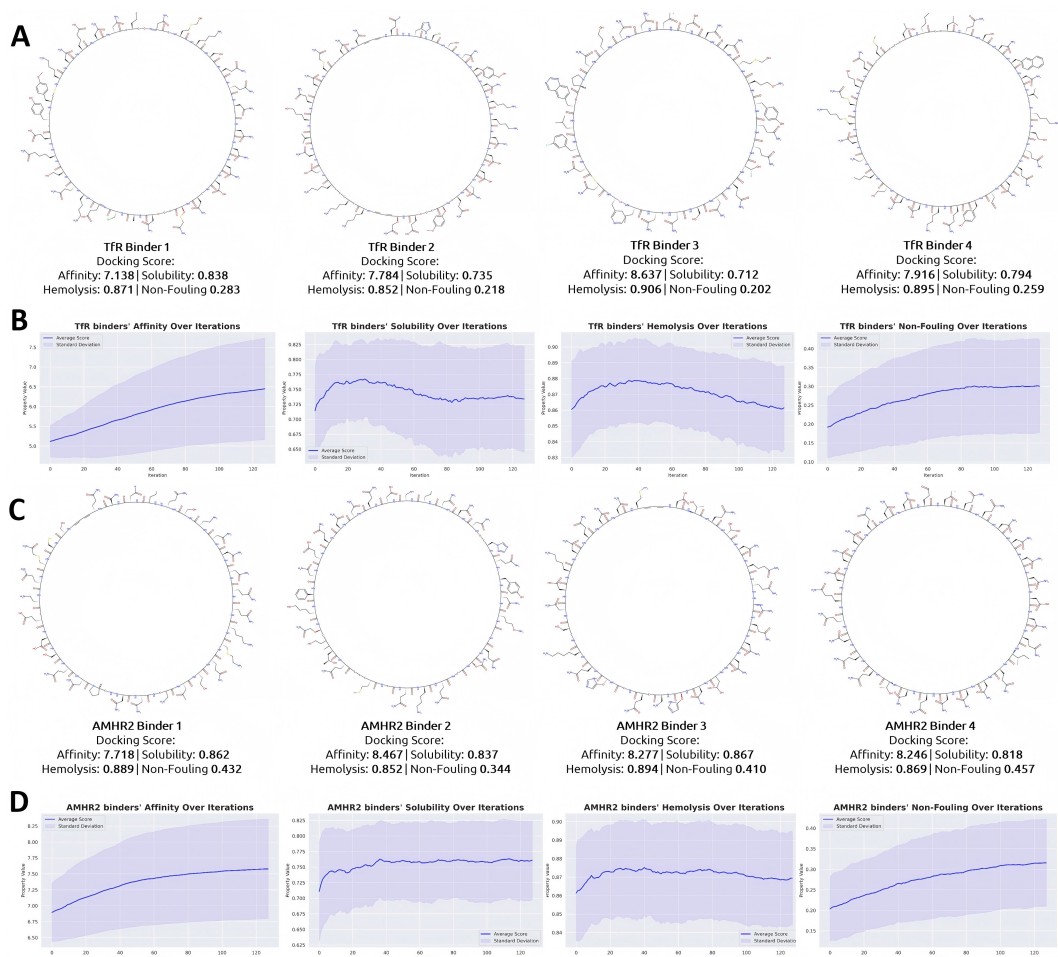

Figure S4: **(A), (C)** Example 2D SMILES structure of AReUReDi-designed peptide binders with four property scores for TfR and AMHR2, respectively. **(B), (D)** Plots showing the mean scores for each property across the number of iterations during AReUReDi's design of binders of length 200 for TfR and AMHR2, respectively.

---

**Algorithm 1** AReUReDi: Annealed Rectified Updates for Refining Discrete Flows

---

1: **Input:** Pre-trained ReDi model $p_t^i(\cdot|x_t)$, objective functions $\tilde{s}_1, \ldots, \tilde{s}_N$, weight vector $\omega \in \Delta^{N-1}$, annealing parameters $\eta_{min}, \eta_{max}$.
2: **Output:** Sequence $x_T$ with multi-objective optimized properties.
3: **Initialize:**
4:     Sample an initial sequence $x_0$ uniformly from the discrete state space $S$
5:     Sample or specify a weight vector $\omega \in \Delta^{N-1}$
6: **for** $t = 0$ to $1$ with step size $h = \frac{1}{T}$ **do**
7:     **Step 1: Annealing and Coordinate Selection**
8:         Update guidance strength: $\eta_t \leftarrow \eta_{min} + (\eta_{max} - \eta_{min})\frac{t}{T-1}$
9:         Select a position $i$ in the sequence to update: $i \sim \text{Uniform}(\{1, \ldots, L\})$
10:     **Step 2: Proposal Generation via Local Balancing**
11:         Let $C_i$ be the set of candidate tokens from $p_t^i(\cdot|x_t)$.
12:         For each candidate token $y \in C_i$:
13:             Compute scalarized reward ratio $r_i(y; x_t)$:

$$r_i(y; x_t) \leftarrow \frac{\exp(\eta_t \min_n \omega_n \tilde{s}_n(x^{(i\leftarrow y)}))}{\exp(\eta_t \min_n \omega_n \tilde{s}_n(x))}$$

14:         Compute unnormalized proposal distribution $\tilde{q}_i(y|x_t)$ using a balancing function $g(\cdot)$:

$$\tilde{q}_i(y|x_t) \leftarrow p_t^i(y|x_t)g(r_i(y; x_t))$$

15:         Normalize to get the final proposal distribution $q_i(y|x_t)$.
16:     **Step 3: Metropolis-Hastings Acceptance**
17:         Sample a candidate token $y^* \sim q_i(\cdot|x_t)$.
18:         Form the proposed state $x_{prop} \leftarrow x^{(i\leftarrow y^*)}$.
19:         Compute acceptance probability $\alpha_i(x, x_{prop})$:

$$\alpha_i(x, x_{prop}) \leftarrow \min\left\{1, \frac{\pi_{\eta_t,\omega}(x_{prop})q_i(x^i|x_{prop})}{\pi_{\eta_t,\omega}(x)q_i(y^*|x)}\right\} \quad \pi_{\eta_t,\omega}(z) \propto p_1(z)\exp(\eta_t \min_n \omega_n \tilde{s}_n(z))$$

20:         With probability $\alpha_i(x, x_{prop})$, accept the proposal: $x \leftarrow x_{prop}$.
21:         Update time: $t \rightarrow t + h$
22: **end for**
23: **Return:** Final sequence $x_1$.

---

# I USE OF LARGE LANGUAGE MODELS (LLMS)

We acknowledge the use of large language models (LLMs) to assist in polishing and editing parts of this manuscript. LLMs were used to refine phrasing, improve clarity, and ensure consistency of style across sections. All technical content, experiments, analyses, and conclusions were developed by the authors, with LLM support limited to language refinement and editorial improvements.

