# OpenReview forum: "AReUReDi: Annealed Rectified Updates for Refining Discrete Flows with Multi-Objective Guidance"
_ICLR.cc/2026/Workshop/LMRL — ICLR 2026 Workshop LMRL Poster_

### Official Review · Reviewer_WShp · 2026-02-18
**Guided sampling extension of Rectified Discrete Flows for multi-objective sequence design**

**Rating:** 7
**Confidence:** 3

**Review:**

**Summary**

The paper proposes Annealed Rectified Updates for Refining Discrete Flows (AReuReDi), an algorithm for guided sampling with Rectified Discrete Flows (ReDi) toward multiple-objective optimality, and demonstrates applications in biological sequence and chemical molecule design. The method constructs a reward-augmented target distribution and uses locally balanced Metropolis-Hastings updates with annealing to bias samples toward Pareto-optimal regions.

**Strengths**
1. Clear motivation for extending Rectified Discrete Flows (ReDi) to multiple-objective settings via sampling from a reward-augmented distribution.
2. Conceptually clean algorithmic formulation.
3. Demonstrates versatility by applying the method to both peptide sequence and SMILES molecule design tasks.
4. Includes thorough ablation studies examining guidance strength, scalarization choices, and comparisons under matched computational budgets.

**Weaknesses**
1. The use of the monotone-accept heuristic in experiments replaces the Metropolis–Hastings acceptance rule and breaks the theoretical stationary distribution guarantees. As a result, the empirical procedure no longer samples from the formally defined reward-tilted distribution, but instead performs greedy reward optimization. While this improves finite-budget performance, it weakens the connection between the theoretical guarantees and empirical evaluation.
2. The method constructs a reward-tilted distribution on top of the data distribution $p_1(x)$ modeled by the pretrained ReDi generative model, but the empirical evaluation focuses on objective predictor scores. It would strengthen the claims to explicitly assess whether the generated samples remain consistent with realistic sequence distributions, particularly given the use of monotone-accept heuristic, which removes formal convergence guarantees to the defined target distribution.
3. The submission is most naturally positioned as a guided generative modeling method for biological sequence design, which aligns with the workshop's data modalities and generative modeling themes. However, while the method leverages a pretrained discrete flow model that implicitly captures meaningful biological structure, the work does not directly analyze or characterize learned representations. Therefore, its connection to the workshop's central theme of representation learning is somewhat indirect.

---

### Official Review · Reviewer_XpMf · 2026-02-24
**Review of AReUReDi**

**Rating:** 7
**Confidence:** 2

**Review:**

**Summary:**

The authors present AReUReDi, a framework for multi-objective optimization (MOO) for discrete sequences. Using Rectified Discrete Flows (ReDi) (which addresses the issue of regular discrete flow matching models assuming each position is independent), the authors focus on enabling MOO under this framework. To do this, they implement 3 strategies: Tchebycheff scalarization (tries to improve performance on the objective which is performing the worst among several objectives) with an annealed guidance strength schedule, locally balanced proposals (updating one position at a time based on how it would impact the reward), and a Metropolis-Hastings update (accepts token based on whether it improves MO score or with random small probability). The authors provide theoretical guarantees for convergence to the Pareto front and empirically show advantages over evolutionary baselines (NSGA-III) and recent diffusion models (PepTune) when generating peptides and SMILES sequences across several therapeutic properties.

**Strengths:**
- Using Rectified Discrete Flows (ReDi) as a backbone is a logical choice for biological sequences. By iteratively straightening the probability paths, the model reduces the factorization error, which leads to more biologically realistic priors during optimization
- The three implemented strategies are simple and well motivated by the mathematical foundation in the appendix, and show substantial empirical improvements in relevant therapeutic properties like half-life, solubility, etc
- There are ample ablation experiments in the appendix to demonstrate how the different strategies come together to motivate the model setup

**Weaknesses:**
- Bond-aware noising schedule - by forcing the model to reconstruct the backbone before the variable side chains, its unclear if the model would be able to correct a backbone choice if it start generates side chains that are incompatible with that scaffold
- Random scan mixture - for longer sequences, a single-coordinate update is a very "local" move. Its possible that the path from a low-affinity sequence to a Pareto-optimal one may require multiple simultaneous changes which may explain why the monotone-accept heuristic is needed. In a 586 token vocab like the SMILES strings the mixing time would be very high with the regular MH update. This seems to indicate the theoretical guarantees are very unlikely to hold in real world application. While this is noted by the authors, I think the text overplays the theory.

---

### Official Review · Reviewer_jg8Z · 2026-02-25
**Multi-objective rectified discrete flows for peptide design**

**Rating:** 6
**Confidence:** 2

**Review:**

## Summary
This paper proposes AReUReDi, a multi-objective optimization framework built on top of Rectified Discrete Flow (ReDi). The method integrates annealed Tchebycheff scalarization, locally balanced proposals and Metropolis–Hastings (MH) updates to bias discrete flow sampling toward Pareto-optimal regions while preserving distributional invariance. The method is applied to wild-type peptide binder design (up to 5 objectives) and chemically modified peptide SMILES generation (4 objectives). The authors claim theoretical guarantees of convergence to the Pareto front, improved trade-off navigation over classical multi-objective optimization (MOO) evolutionary and diffusion baselines and strong in silico performance.

## Strengths
- Clear technical motivation of extending rectified discrete flows to MOO.
- Sound theory.
- Clear improvements over classical MMO algorithms in the experiments.
- Clear biological context of peptide design optimization.

## Weaknesses
- Theory was proven for the unconstrained MH sampler but the experiments used a monotone-accept heuristic that breaks reversibility, so a deeper discussion is needed, i.e. if any of the theory is still applicable, how the heuristic changes the approach.
- Evaluation is based only on predicted properties. If experimental validation is not possible, it would be good to at least add calibration and uncertainty analyses.
- The reported runtime is quite long compared to the baselines. More information on scalability with respect to the sequence length and number of objectives is needed.
- Would be nice to have comparison to more MOO algorithms, e.g. Multi-objective GFlowNets.

## Questions
- How do you ensure that AReUReDi is not exploiting weaknesses in the property predictors? I.e. that maximizing the model objective is related to true biological performance?

---

### Meta-Review · Area_Chair_k8tn · 2026-02-27

**Recommendation:** Accept (Poster)
**Confidence:** 4

**Metareview:**

Accept.

---

### Decision · Program_Chairs · 2026-03-02

**Decision:**

Accept (Poster)

**Comment:**

Please see the meta-review.